# Femtosecond control of phonon dynamics near a magnetic order critical point

O. Yu. Gorobtsov [1,8 ✉], L. Ponet[2,3,8], S. K. K. Patel[4,5], N. Hua[4,5], A. G. Shabalin[4], S. Hrkac[4], J. Wingert[4], D. Cela[4], J. M. Glownia[6], D. Zhu[6], R. Medapalli[5,7], M. Chollet[6], E. E. Fullerton [5], S. Artyukhin [2 ✉], O. G. Shpyrko[4,5] & A. Singer [1 ✉]

The spin-phonon interaction in spin density wave (SDW) systems often determines the free energy landscape that drives the evolution of the system. When a passing energy flux, such as photoexcitation, drives a crystalline system far from equilibrium, the resulting lattice displacement generates transient vibrational states. Manipulating intermediate vibrational states in the vicinity of the critical point, where the SDW order parameter changes dramatically, would then allow dynamical control over functional properties. Here we combine double photoexcitation with an X-ray free-electron laser (XFEL) probe to control and detect the lifetime and magnitude of the intermediate vibrational state near the critical point of the SDW in chromium. We apply Landau theory to identify the mechanism of control as a repeated partial quench and sub picosecond recovery of the SDW. Our results showcase the capabilities to influence and monitor quantum states by combining multiple optical photo-excitations with an XFEL probe. They open new avenues for manipulating and researching the behaviour of photoexcited states in charge and spin order systems near the critical point.

[1] Materials Science and Engineering Department, Cornell University, Ithaca, NY, USA. [2] Central Research Labs, Italian Institute of Technology, Genova, Italy. [3] Scuola Normale Superiore, Pisa, Italy. [4] Department of Physics, University of California, La Jolla, San Diego, CA, USA. [5] Center for Memory and Recording Research, University of California, La Jolla, San Diego, CA, USA. [6] The Linac Coherent Light Source, SLAC National Accelerator Laboratory, Menlo Park, CA, USA. [7] Department of Physics, School of Sciences, National Institute of Technology, Tadepalligudem, Andhra Pradesh, India. [8] These authors contributed equally: O. Yu. Gorobtsov, L. Ponet. ✉email: gorobtsov@cornell.edu; sergey.artyukhin@iit.it; asinger@cornell.edu

If an energy flux passing through the system induces a phase transition far from thermal equilibrium, the pathway of the transition changes significantly, and intermediate states can arise in the material[1,2]. In particular, inhomogeneous excitations produce a lattice displacement in crystalline materials, generating intermediate vibrational states. In materials with a SDW state, vibrational states interact with spin order[3], leading to effects such as a re-emergence of a suppressed SDW[4]. On the other hand, suppressing the intermediate states allows reproducing adiabatic transitions on much faster timescales – an approach termed "shortcuts to adiabaticity" (STA) in quantum technology[5–7]. Manipulation of photoexcited density wave states also has possible applications in fast switching devices[8,9]. Therefore, generating a designed vibrational state in spin systems by enhancing or suppressing spin-phonon interactions opens ways to precise manipulation of spin and lattice systems.

A promising path for altering intermediate states and the energy landscape is to leverage the critical behaviour near a phase transition, where the degree of order in the system changes rapidly with temperature. Elemental chromium serves as an ideal material for studying critical behaviour. In equilibrium, an incommensurate SDW in chromium is accompanied by a charge density wave (CDW) through the electron-phonon interactions and a periodic lattice distortion (PLD) through magnetostriction[3]. The periods of the CDW and the PLD are harmonically related to the period of the fundamental SDW: both have half the SDW period. The high Néel temperatures (311 K in bulk[3], decreased due to dimensional crossover[10] to 290 ± 5 K in the $d = 28$ nm film studied here[11,12]) render the critical behaviour close to the SDW phase transition highly accessible. Thin antiferromagnetic films with SDW states are especially important for devices utilizing spin transport[13,14] and sustain a unidirectional SDW and discrete SDW periods due to pinning at the interfaces[15,16]. The magnetostrictive coupling between SDW and PLD allows for the excitation of acoustic phonons by fully or partially quenching the SDW order parameter throughout the film with a photoexcitation[17–19]. Previous studies have shown that when the incident energy flux at the film is at or below ~2 mJ/cm², the spin order recovers before the acoustic phonon decays[17], transiently enhancing the PLD above its value in equilibrium[2]. The ultrafast recovery of the SDW should enable a repeated quench that acts as a lever to further enhance or suppress the oscillation of the PLD. Additionally, coherent control requires long coherence time of the system, and we specifically exploit long lifetime of the excited acoustic phonon in chromium compared to other materials (e.g. Neugebauer et al.[20]).

High precision measurements of the PLD are necessary to detect and control the changes in the spin and lattice structure effectively. XFELs offer time resolution down to femtoseconds in pump-probe experiments, making them a powerful tool to study PLD dynamics[21]. In thin films, boundary conditions enforce half-integer $(N + 1/2)$ number of PLD periods[12,16,22], corresponding a PLD wave vector to a certain Laue fringe. The X-ray scattering intensity on the Laue fringe is directly proportional to the magnitude of the PLD[16,17,22] (see also Supplementary Note 2), enabling a quantitative measurement of both amplitude and phase of the lattice oscillation. X-ray measurements offer an advantage over recently matured ultrafast electron diffraction:[23,24] the Bragg angles are generally large, and the peaks are narrow, which provides higher peak selectivity and enables measurements of thin films grown epitaxially on a thick crystalline substrate. Moreover, X-rays directly couple to the core electrons, overcoming the challenges of the optical pump-probe methods[8,25,26] where screening effects make an interpretation challenging. In addition, unlike optical reflectivity, X-ray measurements are not hindered by the surface roughness of the film.

## Results and discussion

Here, we use two consecutive laser pulses[7,25,26] to drive the system out of equilibrium; the separation between the pulse arrival times $\tau_1$ and $\tau_2$ modifies the response of the system (e.g. Onozaki et al.[25]). The first ~40 fs laser pulse quenches the electronic and spin order[27] throughout the entire 28 nm Cr film as the optical skin depth in Cr at 800 nm wavelength is ~30 nm[22,28]. The PLD is then released as an acoustic phonon with wavevector normal to the surface and an oscillation period of ~450 fs. In less than a picosecond, the electronic subsystem thermalizes with the lattice below the Néel temperature[27], and the SDW order recovers[29], inducing the recovery of the PLD[17]. The acoustic phonon has a damping time of ~3 ps[17]. If a second laser pulse arrives before the excited state diminishes, it is possible to either further excite, sustain, or completely dampen the excited state (Fig. 1a). Further evolution of the system depends on the phase and amplitude of the phonon in that moment. An X-ray probe pulse of 50-fs duration scatters on the atomic structure of the film at a delay time $t$, and the PLD in the crystal, gives rise to a satellite peak on the Laue fringes of the out-of-plane (002) crystal Bragg peak (Fig. 1a, inset), where the total peak intensity $I(q, t)$ as a function of a wavevector $\boldsymbol{q} = (0, 0, q)$ and time after excitation $t$ (refs. [17,22] and Supplementary Note 2)

$$I(q,t) = I_0 |F_u(q)|^2 \left[ |f(q)|^2 + q A_{\text{PLD}}(t) \sin(\alpha) [f(q)f(q-2Q) - f(q)f(q+2Q)] \right],$$
(1)

where $I_0$ is the normalization constant, $F_u(q)$ is the structure factor of the unit cell, $f(q) = \sin(Nqa/2)/\sin(qa/2)$ describes the Laue fringes, $\alpha = Qa[N - 1] - \phi_0$, PLD is defined as a spatial wave $A_{\text{PLD}}(t)\cos(2Qr_n^0 - \phi_0)$, $r_n^0 = n \cdot a$ are the undistorted atomic positions, $a$ is the lattice constant, $A_{\text{PLD}}(t)$ is the PLD amplitude, $2Q$ is the wave vector of the PLD, and $\phi_0$ defines the offset of the wave. An XFEL pulse 0.2 mm in diameter envelopes multiple SDW domains[30] below the Néel temperature, making the measurement statistical over domains.

Evolution of the PLD amplitude $A_{\text{PLD}}(t)$ as a function of the probe delay $t$ presented in Fig. 1c, d for two different $\tau_2 - \tau_1$ shows that we are indeed able to completely suppress or further enhance the excited state with the second pulse. The phase offset $\phi_0$ is constant and $A_{\text{PLD}}(t)$ is negative when the sign of $A_{\text{PLD}}(t)\cos(2Qr_n^0 - \phi_0)$ is reversed. The transient magnitude of the PLD oscillation (half of the difference between the maximum PLD and the minimum PLD within one period) reaches 125% of the unperturbed lattice distortion (see Fig. 1d after the second pump pulse). The absolute value of the PLD amplitude after both photoexcitations reaches 150%. The variation in PLD is thus six times larger than the total relative drop of 20% in the PLD after 9 ps when the film equilibrates at a new film temperature. An excitation of such a magnitude is reminiscent of impulsive excitation of coherent phonons[31]: the PLD recovers to 80% of the original value long before the acoustic phonon dampens. In the conventional displacive excitation, the total PLD change is about the same as the excitation magnitude[32]. We specifically use the critical behaviour of the order parameter (SDW) in proximity to the critical point to exert control of another coupled mode (PLD). This is in contrast to directly driving the target mode[33].

Figure 2a demonstrates a full map of how the amplitude of the PLD changes with both $\tau_2 - \tau_1$ and the probe delay $t$ for 2 laser pulses of the same fluence $P = 1.5$ mJ/cm². The total laser fluence was chosen to reach the maximum enhancement of the transient PLD amplitude after one pulse laser excitation. The frequency of the acoustic phonon remains largely unchanged, but its amplitude changes periodically with the pump-pump delay. After 9 ps from the last excitation, the PLD stabilizes at a delay-independent level: the lattice and electrons are in the thermal equilibrium and the

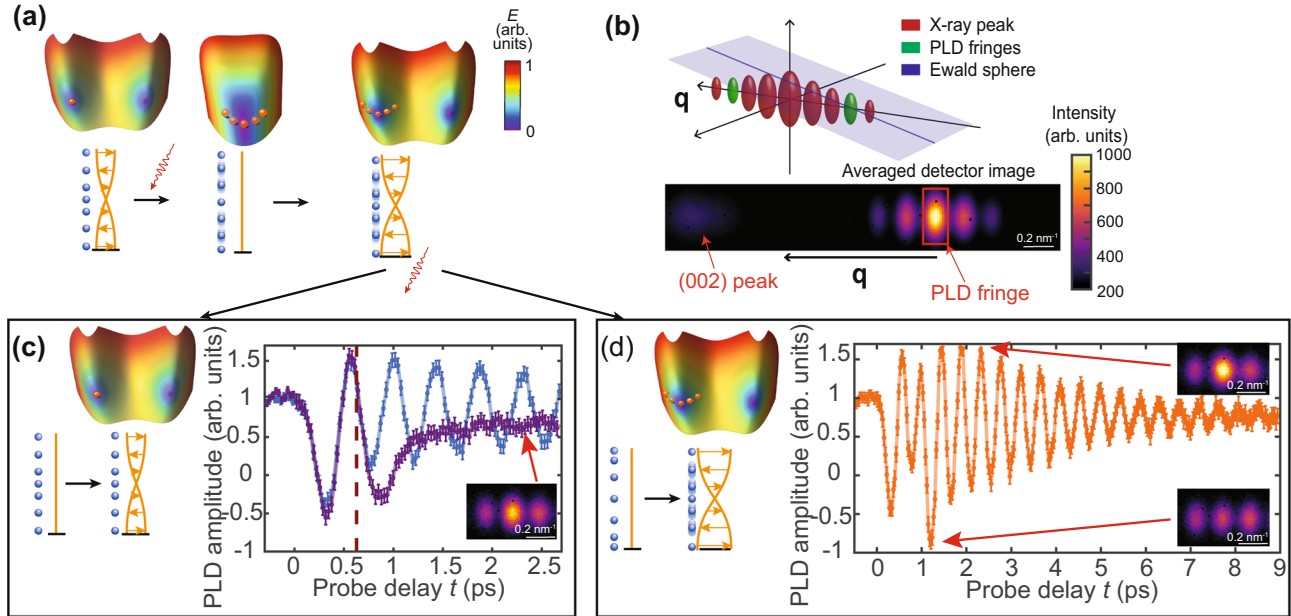

**Fig. 1 Controlled enhancement and destruction of the excited state. a** A laser pulse (red arrow wave) excites the system (blue circles show atomic positions, light orange line shows SDW) by destroying the magnetic order; a second pulse either excites the released phonon further or stops the excitation. Schematic energy ($E$) surfaces (not to scale) show the PLD state as the orange point. **b** Detector image of the X-ray scattering from periodic atomic displacement on a Laue fringe of the main peak from the crystalline film. Here $q = 4\pi/\lambda \cdot \sin(\theta)$ is the scattering vector with $\lambda$ being the x-ray wavelength and $\theta$ the scattering angle. Central peak is weaker than the fringes due to the position of the Ewald sphere. **c, d** Amplitude of the PLD in two extreme control cases. Solid lines connect experimental data points. Vertical bars show uncertainties, estimated as a standard deviation $\sigma_{FEL}$ of the PLD amplitude at $t < 0$ and as $\sigma_{full} = \sqrt{\sigma_{FEL}^2 + \sigma_{ex}^2}$ at $t > 0$, where $\sigma_{ex}$ is a standard deviation of the minimum reached PLD amplitude after the first excitation between different scans with identical fluence $P$. The vertical dashed red line marks the time of second pulse arrival. The incident pulse fluences are $P_1 = P_2 = 1.5$ mJ/cm². **c** Pump–pump delay ($\tau_2 - \tau_1$) 620 fs. Inset shows the PLD Laue fringe after the suppression of the excited state (red arrow points to the time). Colorscale is shared with (**b**).Comparison between the PLD amplitude with a single pulse (blue curve) and two pulses (purple curve) is shown. **d** Pump–pump delay ($\tau_2 - \tau_1$) 845 fs, orange curve Insets show the Laue fringe with the satellite peak at maximum (top) and minimum (bottom) PLD values (red arrow points to the time). Colorscale is shared with **b**.

subsequent heat transport to the substrate occurs over ~1 ns timescale[22]. When $|\tau_2 - \tau_1|$ is shorter than the time of the SDW recovery, we observe an ~2 times weaker excited phonon (Fig. 2a), as the second pulse merely increases the overall temperature of the system without displacive excitation[32]. Remarkably, the second pulse can be significantly weaker, but the effect will still remain: Fig. 2b shows the full map when the second pulse is weakened by a factor of 2. Another important observation is the saturation effect: at a fluence of ~10 mJ/cm² or higher, the overall temperature of the system rises above $T_N$, and the system is indifferent to the second pulse arrival time $\tau_2$: compare the "enhanced" and "suppressed" phonons in Fig. 2c, which are virtually the same. At the fluences of and higher than 10 mJ/cm², the magnetic order is destroyed after the first pulse and does not recover; therefore, the second pulse has a negligible effect on the phonon dynamics. Interestingly, we do not observe a prolonged order recovery on the scale of several picoseconds associated with the removal of topological defects[2] (see Methods section).

Now we rationalize the results with a non-equilibrium phenomenological model based on a combination of Landau-type theory and two temperature model. The laser pulse with power $Q_{ph}(t)$ acts to increase the energy of the electronic subsystem, which thermalizes on 100 fs timescale, and is therefore thermal on the ps timescales of our interest. The SDW is a part of the electronic subsystem and is thus characterized by the same temperature $T_L$. On a longer timescale the electronic subsystem cools by transferring the energy to the lattice phonon bath (excluding the PLD mode) with temperature $T_b$, which thermalizes through anharmonic phonon interactions. The temperature evolution of the electronic system absorbing photons and exchanging heat

with the bath is described by (see refs. [27,34,35])

$$C_L \dot{T}_L = -k\left[T_L(t) - T_b(t)\right] + Q_{ph}(t), \quad (2)$$

$$C_b \dot{T}_b = -k\left[T_b(t) - T_L(t)\right]. \quad (3)$$

Here $C_L = c_L T_L, C_b$ are the heat capacities of the electronic subsystem and the bath, $c_L$ is proportional to the electronic Sommerfeld coefficient, $k$ is the thermal coupling, and $Q_{ph}(t) = \frac{A}{\xi\sqrt{2\pi}} e^{-\frac{1}{2}\frac{(t-t_0)^2}{\xi^2}}$ is the heat injected by the photon pulses, $\xi$ is the pulse time constant, and $A$ the pulse strength.

The changes in the $T_L$ induced by the laser pulses affect the SDW amplitude as described by a Landau-type theory[36] with order parameters $L$ and y denoting the amplitudes of the SDW and the PLD, related to the Fourier component of the spin density at the SDW wave vector $q$, $L = S_q$ and to the normalized acoustic phonon amplitude y $= u_{2q}/y_0$, respectively (Fig. 3a). The essential energetics of the interacting PLD and SDW can be captured by the Landau free energy,

$$F(L,y) = \frac{\alpha}{2}\left(T_L - T_N\right)L^2 + \frac{\beta}{4}L^4 - gL^2y + \frac{\rho y_0^2 \omega_0^2}{2}y^2 + \frac{b}{4}y^4, \quad (4)$$

where $T_L$ is the temperature of the spin subsystem, $T_N$ is the Néel temperature; the terms with $\alpha$ and $\beta$ describe the double-well potential for the SDW amplitude $L$, and the phonon with amplitude y is characterized by the displacement amplitude $y_0$, density $\rho$, frequency $\omega_0$ and anharmonicity $b$. The lowest-order interaction term between SDW and PLD with the coupling constant $g$ (akin to refs. [37,38]) contains the time reversal-odd $L$

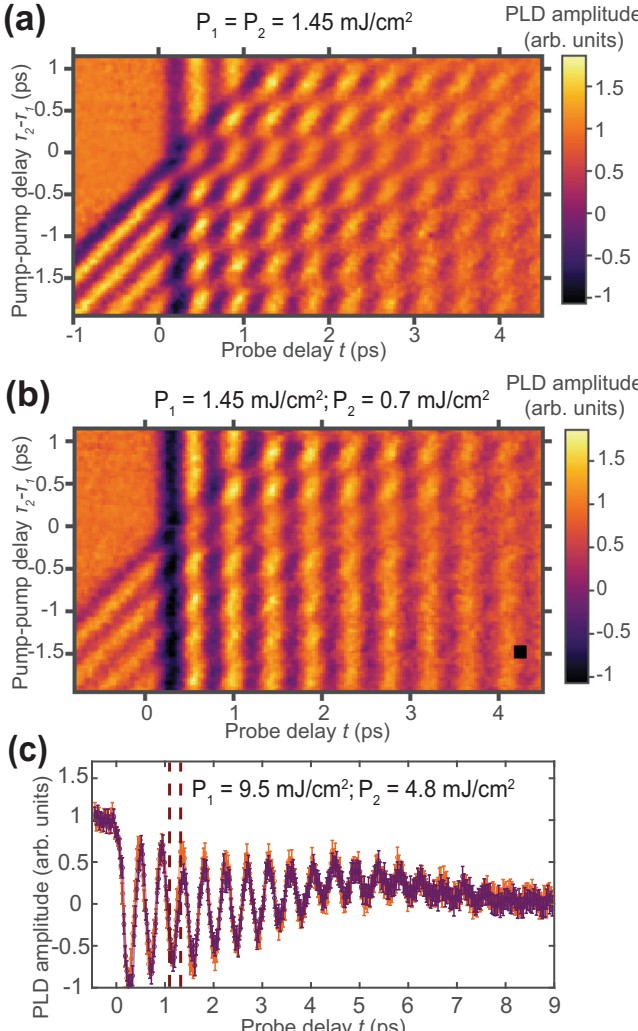

**Fig. 2 Experimental maps of the excited state depending on the probe and pump delay. a** Map of the PLD amplitude before and after excitation by two pulses with fluences $P_1 = P_2 = 1.5$ mJ/cm$^2$. **b** Map of the PLD amplitude before and after excitation by two pulses with fluences $P_1 = 1.5$ mJ/cm$^2$, $P_2 = 0.7$ mJ/cm$^2$. **c** Magnitude of the PLD in "enhancement" and "suppression" conditions with laser fluences $P_1 = 9.5$ mJ/cm$^2$ and $P_2 = 4.8$ mJ/cm$^2$. Solid lines connect experimental data points . Vertical bars show uncertainties, estimated as a standard deviation $\sigma_{FEL}$ of the PLD amplitude at $t < 0$ and as $\sigma_{full} = \sqrt{\sigma_{FEL}^2 + \sigma_{ex}^2}$ at $t > 0$, where $\sigma_{ex}$ is a standard deviation of the minimum reached PLD amplitude after the first excitation between different scans with identical fluence $P$. The dashed red lines mark the times of second pulse arrival. Data for $\tau_2 - \tau_1$ 1295 fs (orange line), $\tau_2 - \tau_1$ 106 fs (purple line) are shown.

squared, for the energy to be time reversal-even, and describes the exchange striction. It leads to the force on the phonon mode $f = -\frac{\partial F}{\partial y} = gL^2$ which drives the acoustic phonon amplitude $y$, as described by an oscillator equation, that follows from Euler-Lagrange equations with the potential of Eq. (4),

$$\rho y_0^2 \ddot{y} = gL^2 - \rho y_0^2 \omega_0^2 y - by^3 - \gamma \dot{y}, \qquad (5)$$

with damping $\gamma$ and dots designating time derivatives. We fit Eqs. (2–4) to the measured normalized PLD amplitude $y(t) = A_{PLD}(t)/A_{PLD}(0)$, taking $y_0 = 0.5 \times 10^{-12} m^{21}$ and $\rho = 7150 \frac{kg}{m^3}$ and $L(t = 0) = 1$, to obtain the model parameters in Eq. (4) as $\alpha = 5.1 \times 10^7 \frac{J}{Km^3}, \beta = 9.0 \times 10^9 \frac{J}{m^3}, g = 3.46 \times 10^5 \frac{J}{m^3}, \frac{\omega_0}{2\pi} = 2.18$ THz, $b = 1.16 \times 10^4 \frac{J}{m^3}, \gamma = 1.37 \times 10^{-9} \frac{J \cdot s}{m^3}$. Parameters

in Eqs. (2) and (3) are defined to a constant multiplier (except for the pulse duration $\xi = 40 fs$) by fitting. The constant can be fixed based on literature value of $C_b = 3.23 \times 10^6 \frac{J}{m^3 K}$[35], obtaining $k = 1.25 \times 10^{18} \frac{W}{m^3 K}$, $c_L = 1.91 \times 10^3 \frac{J}{m^3 K^2}$, mean value of $A = 8 \times 10^7 \frac{J}{m^3}$.

The magnitudes of $L$ and $y$ are dimensionless in the model, however the measured displacement amplitude $y_0$[16], the measured frequency $\omega_0$ and the material density $\rho$ fix the energy scale in Eqs. (4) and (5), while the bath/lattice heat capacity $C_b$ of Cr[35] fixes the parameter values in Eqs. (2) and (3). The anharmonic term gives a small variation in frequency between excitations ($b\rho y_0^2 \omega_0^2 \sim 0.03$). Values for heat capacities obtained from fitting are within the expected range: the ratio between the lattice and electronic heat capacities during the excitation ($C_b/c_L T \sim 7$) is comparable to that extracted from previous experiments[29] (larger $c_L$ values than in bulk[35] can be explained by disorder in the film[39]), as is the value of thermal coupling constant $k$[35]. Absorption at 800 nm wavelength and 30° angle can be estimated based on measured complex refraction index $n \approx 3 + 1.1i$ for Cr thin films[40] as ~20%, while the obtained value of $A$ gives absorption of $Ad/P \sim 15\%$, which is remarkably close considering possible differences due to film roughness. With these fitted parameters, our theoretical model provides an excellent quantitative agreement with the experiment, as Fig. 3d, e demonstrate.

As $L$ is partially melted by the laser pulses, the force $f$ changes and the acoustic phonon is excited. Below $T_N$, the non-zero SDW varies as $L \sim \sqrt{T_N - T_L}$ within the mean-field approximation, as can be found by minimizing the free energy (4) with respect to $L$. Through the exchange striction $g$ this leads to a non-zero force $gL^2$ on the PLD $y$, shifting its minimum away from zero. When the laser pulse quenches the SDW, $L$ rapidly decreases and then returns close to its original equilibrium value within ~0.5 ps as the electronic system cools. Through exchange striction, these changes of $L$ launch an oscillation of $y$. The SDW amplitude $L$ saturates at temperatures well below the Néel temperature ($T \ll T_N$) but is strongly sensitive to $T$ near the phase transition due to the square root power law. Therefore, the oscillation is induced most effectively in the temperature range close to the critical point.

We now discuss the behaviour of the coupled modes $L$ and $y$ displayed in Fig. 3b, c. The maximum momentum transfer can be achieved when the SDW order is first fully melted $gL^2 = 0$, shifting the minimum of the PLD potential to zero. The potential should then remain stationary until $y$ crosses through the minimum, and then move back towards its original equilibrium position, to achieve the maximum oscillation amplitude of the PLD mode. Since the SDW cools rapidly it is thus beneficial to heat the SDW a bit above $T_N$, just so the PLD mode can catch up and pass all the way to the minimum at zero (the condition achieved, according to the model, in Fig. 2a, but not Fig. 2b). However, after this process, until the SDW order is restored at least partially, no further momentum can be transferred, leading to the horizontal stripe with a relatively low oscillation amplitude around zero pump-pump delay ($T_1 = T_2$) when both pump pulses come almost simultaneously (Fig. 3d and cf. experimental Fig. 2a, b). If the SDW order is partially melted, and the second pulse hits after it is restored, y gains extra momentum. Depending on the timing of the second pulse with respect to the first one, this extra momentum can be along or opposite to that already present, therefore increasing or decreasing the oscillation amplitude, as seen from experiments in Figs. 1 and 2. If the second pulse decreases the oscillation amplitude, the excess energy of the phonon transfers to the SDW through coupling between SDW and PLD and then dissipates into the phonon bath. The excess energy of the phonon is negligible compared with the energy influx from the laser excitation (experimentally, the long-time damping of the phonon does not lead to a change in the average

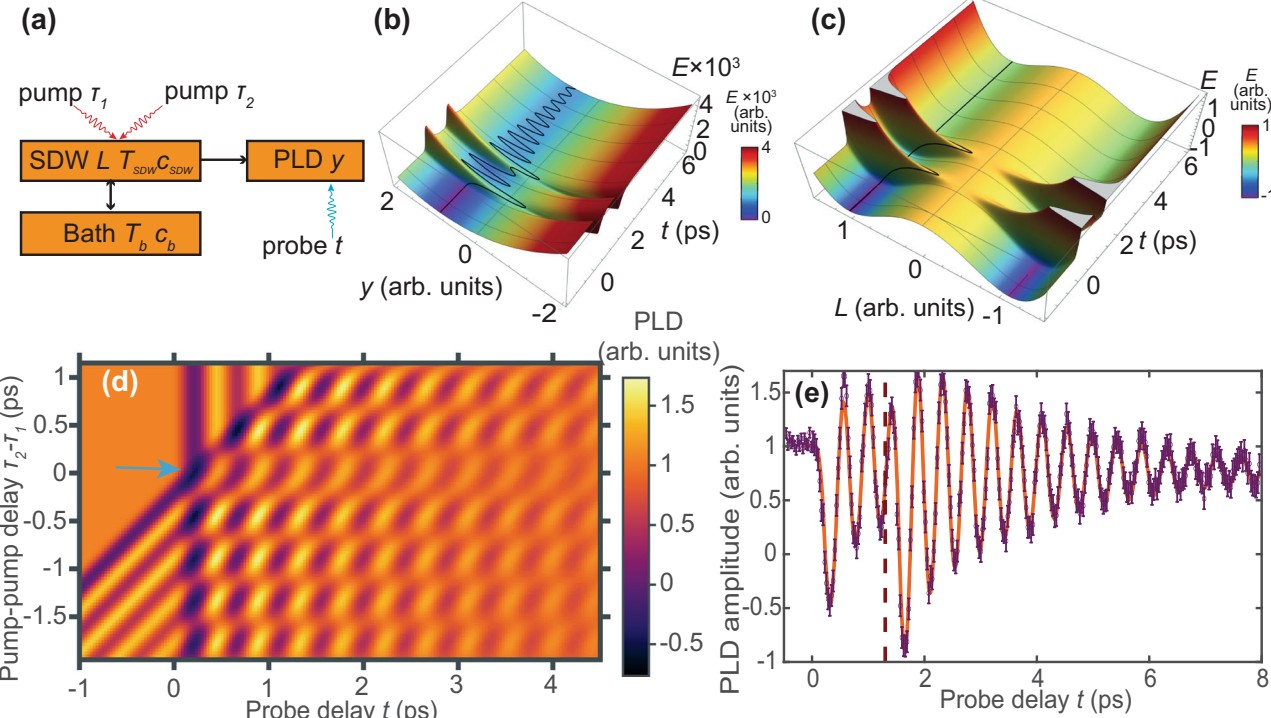

**Fig. 3 Theoretical model. a** Schematic representation of the model. Orange blocks represent different parts of the model: SDW, PLD, and a thermal bath. **b, c** Schematic evolutions of $y$ and $L$ on their respective potential surfaces. Features of the potential surfaces $E(y,t)$ and $E(L,t)$ are exaggerated, and the color accentuates the surface shape, for illustration purposes. **d** Map of the simulated amplitude of the acoustic mode, depending on pump–pump, and pump–probe delay. The faint row, highlighted by the blue arrow, at 0 ps pump–pump delay demonstrates the weaker excitation by a single high-fluence pulse. Higher amplitudes are achieved at higher pump–pump delay times. **e** Typical example of the fit achieved by the model. Purple line is the experimental data, vertical bars show uncertainties, estimated as a standard deviation $\sigma_{FEL}$ of the PLD amplitude at $t < 0$ and as $\sigma_{full} = \sqrt{\sigma_{FEL}^2 + \sigma_{ex}^2}$ at $t > 0$, where $\sigma_{ex}$ is a standard deviation of the minimum reached PLD amplitude after the first excitation between different scans with identical fluence $P$. Orange line is the theoretical model. The vertical red dashed line marks the time of second pulse arrival.

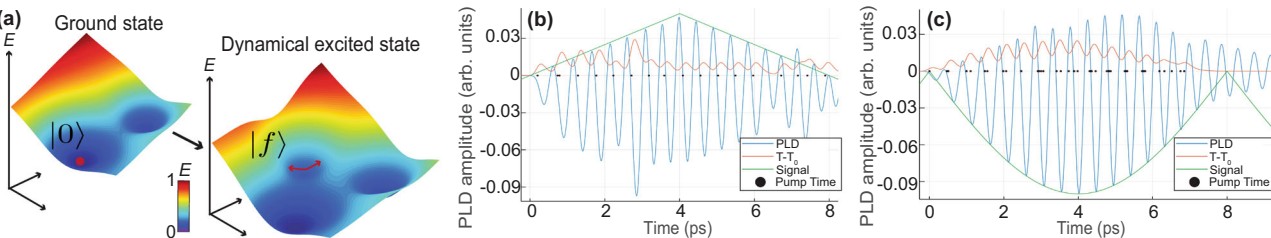

**Fig. 4 Simulation of a pulse train excitation. a** Phenomenology for advanced control, suspending the system starting in a ground state |0> in an excited state |f>. Energy ($E$) surface shown. **b, c** Optimal control of the oscillation, guiding the envelope of the PLD signal (blue lines) along the desired (green lines) **b** triangular or **c** sinusoidal curve by applying a train of 15 pulses with the same fluence, timed at the moments indicated by black circles on the horizontal axis. The deviation from the initial temperature is plotted in orange. The simulations used the model parameters determined from the experiment.

PLD value[8]). To maximize the phonon amplitude, multiple pulses, spaced at optimal delays, are essential.

The excellent agreement between theory and data allows us to speculate on the broader control possibilities based on the theoretical parameters extracted from our experiments. By further splitting the laser pulse into a longer pulse train, a further control over a sustained excited state would be possible (Fig. 4), similar to optical control of molecular motion with long pulse trains[41]. Simulations performed for Fig. 4b, c using our experimental model demonstrate a possibility to drive the phonon to follow the desired trajectory, for example with sawtooth (Fig. 4b) and sinusoidal (Fig. 4c) oscillation envelopes. We select a low enough fluence to avoid excessive heating of the material (dynamic equilibrium: all heat brought in can be removed by the cooling

system). The limiting factors for reaching higher oscillation amplitudes are finite pump pulses, cooling rate, damping and heat capacity $C_b$. Furthermore, the starting temperature imposes an upper limit on the maximum force per pulse, achieved when SDW is melted completely, and the oscillation frequency limits the time window wherein the force is effective in increasing oscillation amplitude. Careful balance between these considerations determines the optimal pulse train for a given envelope. In practice, we fix the pulse fluence and optimize the pulse timings one period at a time, allowing for multiple pulses (15 in this simulation) during one period. Fixed pulse fluence and variable timings mimic experimental capabilities. In Cr, the intensity of an individual pulse in such a train would have to be kept below ~0.1 mJ/cm² so that the total deposited heat dissipates

into the substrate and the equilibrium temperature does not rise above $T_N$.

This work establishes an FEL X-ray probe combined with multiple optical photoexcitations in the vicinity of a critical point as a path to rational design of ultrafast control over photoexcited vibrational states and better physical understanding of underlying processes. The ultrafast X-ray probe serves as a high precision feedback to the optical double pump setup, enhancing possibilities for control. The precision of PLD measurements lets us develop a quantitative model as a guide to future developments. Our results pave a way to ultrafast driving of complex spin and charge ordered systems, revealing details of electron–phonon and spin–phonon coupling, especially near criticality.

## Methods

**Film preparation**. The thin chromium film was deposited onto the single-crystal MgO(001) substrate using DC magnetron sputtering at a substrate temperature of 500 °C and annealed for 1 h at 800 °C. The growth process was optimized to yield both a smooth surface and good crystal quality of the sample. The film thickness was determined to be 28 nm by x-ray diffraction. The Néel temperature of a thin film is reduced to 290 ± 5 K.

**Experimental parameters**. The pump-probe experiment was carried out at the XPP instrument of the LCLS with an x-ray photon energy of 8.9 keV, selected by the (111) diffraction of a diamond crystal monochromator. The x-ray polarization is horizontal, and the scattering geometry is vertical. X-ray diffraction in the vicinity of the specular out of plane (002) Bragg peak ($2\theta = 60$ degrees) from each pulse was recorded by an area detector (CS140k) with a repetition rate of 120 Hz. The film was cooled down to 115 k with a cryojet. Due to the mosaic spread of the crystal in the film plane, a number of Laue oscillations are observed on the area detector simultaneously. About 100 pulses were recorded for each time delay (50 fs steps in the time traces). For each time delay, the intensity was dark noise corrected and normalized by the intensity measured in the region of the area detector where Laue oscillations were absent. The sample was excited by optical (800 nm, 40-fs), p-polarized laser pulses propagating nearly collinear with the x-ray pulses. The incident fluence on the sample was controlled by an angle of an optical waveplate between polarizers and measured with a photodiode. The final temporal resolution was estimated to be 80 fs. The spot sizes (full width at half maximum) of the optical and x-ray pulses were 0.46 mm (H) × 0.56 mm (V) and 0.2 mm (H) × 0.2 mm (V), respectively. The combined power of both optical laser pulses in measurements without filters was 2.9 mJ/cm$^2$ in standard runs and 19 mJ/cm$^2$ in high power runs. A 0.3 OD ND filter with a transmittance of 0.5 was used to weaken the second laser pulse in the relevant measurements.

## Data availability

Raw data were generated at the Linac Coherent Light Source (LCLS), SLAC National Accelerator Laboratory large-scale facility. Derived data supporting the findings of this study are available from the corresponding author upon request. Source data used for figures are available in the Open Science Framework repository at https://osf.io/rgf8h/?view_only=c7a8cd7244044b638c748c0b3b8ce106.

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

## Acknowledgements

We thank Guru Bahadur Singh Khalsa for useful discussions. The work was supported by U.S. Department of Energy, Office of Science, Office of Basic Energy Sciences, under Contracts No. DE-SC0019414 (ultrafast x-ray data analysis and interpretation O.Yu.G. and A.S.), DE-SC0001805 (ultrafast x-ray scattering experiments, A.S., N.H., A.G.S., S.H., J.W., D.C., and O.G.S.) and No. DE-SC0018237 (thin films synthesis and characterization, ultrafast x-ray scattering, S.K.K.P. and E.E.F.). Use of the Linac Coherent Light Source (LCLS), SLAC National Accelerator Laboratory, is supported by the U.S. Department of Energy, Office of Science, Office of Basic Energy Sciences under Contract No. DE-AC02-76SF00515.

## Author contributions

A.S., O.G.S., and E.E.F. planned the project; A.S., S.K.K.P., A.G.S., N.H., S.H., R.M, J.W., D.C., J.M.G, D.Z., M.C. performed the pump-probe x-ray measurements; O.Yu.G. performed data analysis and interpretation of the results; L.P. and S.A. worked on the theory; S.K.K.P. and E.E.F. grew samples; and O.Yu.G., L.P., S.A., and A.S. wrote the paper. All authors contributed to discussions and gave comments on the manuscript.

## Competing interests

The authors declare no competing interests.
