## [Peer Review File · Nature Communications]

REVIEWER COMMENTS

Reviewer #1 (Remarks to the Author):

Hereby you will find my report of the manuscript by O. Yu. Gorobtsov and co-authors entitled "Femtosecond control of phonon dynamics near a magnetic order critical point".

The authors report a study of the dynamical interplay of the spin, electronic and vibrational degrees of freedom in Cr films. The main experiments reported consists of a perturbation scheme based on two consecutive photoexcitation and a probe of the periodic lattice distortion by ultrashort X-ray. The authors show that is possible to coherently control the average lattice distortion by appropriately choosing the relative times between the two pumps the second pulse arrives before the excited state diminishes. By exploring part of the parameter space of the photo-excitation, i.e. the relative delay between the pulses at play and their intensity, the authors argue that, when the delay between the two excitations is shorter than the time of the SDW recovery, the second pulse has a weaker effect on the system as the overall temperature of the system rises above the Neel temperature and the system is indifferent to the second pulse arrival time. They propose a model for interpreting the interplay between the phonon dynamic and the magnetic order based on Ginzburg Landau theory describing the interaction between magnetic and vibrational degrees of freedom with two order parameters describing the acoustic phonon amplitude and the spin density wave.

I think the manuscript reports data which represent a high quality example of coherent vibrational control measured with x-ray pulses, which deserved to be published in Nature comm. Further the theoretical interpretation is a minimal model for the effect which gives a sound description of the data.

The only major remark I have for the authors regards the strong focus of the manuscript on the role of the spin density wave in the excitation mechanism for the coherent lattice displacement which in my opinion remains inconclusive. What is the smoking gun for the involvement of the spin density wave order? Wouldn't a more conventional displacive mechanism driven by the electronic temperature subject to large dissipations and diffusion (in the electronic channel) give a very similar phenomenology? Can the author give more details on how the specific lattice distortion measured is coupled to the spin density wave?

Further, regarding in the example of coherent control reported in fig. 1, where does the energy go in the case of a complete quench of the lattice vibration? I guess it cannot be transferred back to the second pump pulse as it is commonly done in ISRS in transparent material, and I find it a bit counterintuitive that the energy could be entirely stored in the SDW as the model suggests. Can the author provide more evidence for this?

I think the manuscript falls short in giving a convincing argument beyond plausibility and would greatly benefit of a more elaborated discussion regarding this. In spite of this short comes, I think the data are of good quality, they show a clear example of vibrational coherent control which deserve to be published once the point raised is taken into consideration and properly discussed.

I have also some other minor input which could help the authors in making their manuscript more accessible:

- What do they mean by "second Harmonic periodic lattice distortion"? I find this wording a bit confusing.

Reviewer #2 (Remarks to the Author):

In the present manuscript Gorobtsov et al discuss an X-ray diffraction experiment on Cr films. The experiment is done in a pump-probe fashion, in which the novelty is a double-pulse excitation prior to the probe pulse, and the expansion of this toward shaping coherent lattice dynamics. While the ideas are interesting and the simulations appear to reproduce experiments, the manuscript lacks significant amounts of even basic information required for reasonably understanding the work. This severely limits the ability to pass reasonable judgement on several claims and conclusions made in the text, and also on the execution of the work itself. The manuscript needs quite significant alteration before it may rise to a reasonable level, let alone to that of Nature Communications.

Also, a metaphorical elephant in the room is the striking resemblance to an existing work with the same novelty and similar time scales (double-pulse ultrafast X-ray diffraction experiment on a CDW system) in Neugebauer et al, Phys. Rev. B 99, 220302(R) (2019) (This reviewer is not a co-author of that work)

As mentioned above, due to the lack of information about the basic data analysis done, I have limited ability to assess the quality of several physical conclusions and statements made, Nevertheless, in the following I address specific issues, starting from the most significant. Without several of these, this work cannot be even properly peer-reviewed.

1. This is a diffraction experiment, and two PLD reflections ("PLD fringes" in Fig. 1a) are supposedly probed. However, there is no explanation of how the authors got to their observable "PLD amplitude" from the intensities they observe. Furthermore, the only raw data shown are series of fringes in 3 insets of Fig 1 which are not clearly labelled. Which of these spots is used? How is it used to reach the PLD amplitude shown in the other figures? Is there a structure factor calculation behind this? If not, what is done and how is this justified? Such steps should be clearly spelled out, especially in the methods, if this work is to be believed and its quality assessed.

2. There are no error bars associated with any of the data. With 100 pulses per delay and 120Hz, the statistics behind these data should be clearly addressed. Without this how would a reader assess/believe the confidence in such data?

3. Fluences – how are the Fluences calculated? Is the similar spots of the pump and the probe taken into account? Are these absorbed or incident fluences? If these are absorbed fluences, how does the reflection geometry affect this calculation? What is the penetration of the pump and how was this calculated? Does this account for reflectivity? How? Why do you write “>” on line 118?

4. Several, even basic, experimental details are missing

a) At what temperature was this experiment done?

b) What X-ray polarization was used? Since you allude to a diamond polarization analyzer, what polarization channel was selected? (I presume sigma-sigma)

c) There is little account of the experimental setup. Was this in reflection geometry? (assumed from the (200) spectral reflection used).

d) Sample – how do you know TN is lower than bulk? Where did the number 290K come from?

5. The authors discuss the temperature of the system and subsystems with respect to TN, eg. in line 118. How is this known? Are Debye-waller dynamics collected and analyzed for assessing lattice temperature?

6. There is some basic confusion surrounding the justification of the simulations. The entire experiment deals with a lattice which is not in a thermal state because of the excited phonons (it cannot be described by Bose-Einstein statistics). However, the simulations deal with temperatures. How is this justified? This should be addressed in detail.

7. Eq. 1 – what is F a function of? Is T_{SDW} the same as T in line 151? The parameters are mentioned in line 134, but what are their values? How do they relate to literature? Which of these are fit parameters? In lines 325 and 326 the fit results for several parameters are given, with no context. Their values and meaning should be addressed/justified on physical grounds, otherwise this model might as well just be non-physical. There are not even units assigned.

8. Theoretical model: what is capital Phi? (line 313) It appears very central to the model. Based on the sketch in 3a I assume it relates to the SDW. How does it relate to L?

9. "g" in the model – what is this parameter based on? Does it relate to exchange striction in the system? Is there previous literature on this? Most likely several decades old.

10. The two-temperature model (TTM) – There is significant context missing. There is a clear lack of detail and justification surrounding how TTM relates to Eq. 1. To my understanding, this model is coupling the SDW to part of the lattice. Its purpose is to describe heat injection from the laser. But it appears to assume that the laser couples to the SDW. Is this true? How is this justified and what are the limits of this assumption? What is the SDW coupled to? A bath of all phonons that are not coherent? Why is this justified? Please consider existing works on the non-thermal lattice even in simple metals such as Al and Sb (PRB B 95, 054302 (2017) and PRX 6, 021003 (2016), again: not a co-author), or momentum-resolved works by M. Trigo at LCLS. Lastly, does this relate to the TTM analysis done on Cr in ref 26? Also, how were the heat capacities of the subsystems computed? Are they even physical?

11. The last point in this work is a demonstration of engineering the dynamics based on the model. While these are nice ideas, there is an absolute lack of information on this, just fancy figures. Please present details that explain and justify what is happening here.

12. The discussion begins (line 176) with a statement that suggests that this work establishes an XFEL with pump pulses as a promising route to control. I do not understand this statement. Do the authors suggest that devices will have small XFELS in them? Surely this statement can be substituted with a more reasonable statement on the relevance of this work.

13. When discussing details in fig. 2a,b and 3d, it would help to clearly define the cases that are discussed in a point-by-point basis. As these panels are rich in detail and very similar, the current descriptions are hard to follow.

14. Why are the Laue fringes commensurate with the CDW peak's q vector?

15. Lines 119-120 – compare 2c top and bottom – what do we see? What is the relevant difference? This is not clear.

16. Line 120 – magnetic order is destroyed and does not recover – how do you know this?

17. Lines 142 and 151 – the mathematical relations, how were they derived?

18. Figures:

a- the figures and panels are not cited in order.

b- Fig 1a – the shape of this colorful potential is not clear, and the colorscale is shown with no units or numerical scale.

c- Fig 1a inset is not clear. What are the indicated directions and what is q ?

d- Fig 1 – there are no labels or units around the images of diffraction peaks. Not even a color scale

e- The lower inset of 1b looks like a convincing piece of this work. I would suggest to show it clearly. Also please indicate the arrival of the 2nd pump in a similar way to the other delay scans.

f- Fig2a,b – differences between these two panels are not clear. A more suitable color scheme or annotation for guidance would be helpful.

g- Fig. 3b,c – what are the conditions of this simulation?

h- Figure 4b,c - are the black dots supposed to be the red dot?

19. This work is about Chromium, an important material that exhibits an SDW, inducing a PLD, and an associated CDW. This is a relatively unique system, or at least a very well known one. But the name Chromium is not in the title nor the abstract. I would suggest to change this. Also, why is Ref. 5 in a abstract in the context of non-equilibrium?

Lastly, I wish everyone associated with this work and their families health and security during these troubling times.

Reviewer 1

R1P1: I think the manuscript reports data which represent a high quality example of coherent vibrational control measured with x-ray pulses, which deserved to be published in Nature comm. Further the theoretical interpretation is a minimal model for the effect which gives a sound description of the data.

We thank the Reviewer for supporting the publication!

R1P2: The only major remark I have for the authors regards the strong focus of the manuscript on the role of the spin density wave in the excitation mechanism for the coherent lattice displacement which in my opinion remains inconclusive. What is the smoking gun for the involvement of the spin density wave order? Wouldn't a more conventional displacive mechanism driven by the electronic temperature subject to large dissipations and diffusion (in the electronic channel) give a very similar phenomenology?

The reviewer is correct in pointing out that in general, the SDW mechanism is not the only possible excitation mechanism in Cr. In our experiment, the SDW is the primary order parameter, and the PLD is caused by it. The ultrafast quench and reordering of the SDW after a photoexcitation has been observed by tr-ARPES in Cr (Ref. [31] in the revised manuscript). In addition, the behavior we observe in the experiment relies specifically on the proximity to the magnetic order critical point. In this paper, we specifically exploit the fast changes of the order parameter near the critical point to produce changes that are significantly stronger than with a conventional displacive mechanism.

We modified the text on page 3 starting at line 89 to make the statements clearer:

“The first 40 fs laser pulse quenches the electronic and spin order [29] throughout the entire 28 nm Cr film as the optical skin depth in Cr at 800 nm wavelength is approximately 30 nm [30] [25]. The PLD is then released as an acoustic phonon with wavevector normal to the surface and an oscillation period of approximately 450 fs. In less than a picosecond, the electronic subsystem thermalizes with the lattice below the Néel temperature [29], and the SDW order recovers [31], inducing the recovery of the PLD [8].”

And on page 3 starting at line 110

“An excitation of such a magnitude is reminiscent of impulsive excitation of coherent phonons [33]: the PLD recovers to 80% of the original value long before the acoustic phonon starts to dampen. In the conventional displacive excitation, the total PLD change is about the same as the excitation magnitude [34]. We specifically use the critical behaviour of the order parameter in proximity to the critical point to induce significant changes without fully destroying the SDW.”

R1P3: Can the author give more details on how the specific lattice distortion measured is coupled to the spin density wave?

In equilibrium the PLD is coupled to SDW through magnetostriction, with displacement direction parallel to the SDW wave vector Q . This PLD is the second harmonic of the SDW: the period of the PLD is twice smaller than the period of SDW (since the PLD is not sensitive to the specific spin direction, see for schemes Ref. 5 in the text, (Fawcett, 1988)). The antinodes of the PLD coincide with the antinodes and nodes of the SDW. We added a scheme in the Supplementary Material (Fig. S1). Out of equilibrium, the dynamics of the SDW measured in tr-ARPES after a single photoexcitation are on the same timescale as the dynamics of the PLD measured here, suggesting their persistent coupling. We clarified in the text on page 1, starting at line 53:

“Elemental chromium serves as an ideal material for studying critical behaviour. In equilibrium, an incommensurate SDW in chromium is accompanied by a charge density wave (CDW) through the electron-phonon interactions and a periodic lattice distortion (PLD) through magnetostriction [2]. The periods of the CDW and the PLD are harmonically related to the period of the fundamental SDW: both have double the SDW period. The high Néel temperatures (311 K in bulk [2], decreased due to dimensional crossover [15] to 290 ± 5 K in the 28-nm film studied here [16] [17]) render the critical behaviour close to the SDW phase transition highly accessible.”

R1P4: “Further, regarding in the example of coherent control reported in fig. 1, where does the energy go in the case of a complete quench of the lattice vibration? I guess it cannot be transferred back to the second pump pulse as it is commonly done in ISRS in transparent material, and I find it a bit counterintuitive that the energy could be entirely stored in the SDW as the model suggests. Can the author provide more evidence for this?”

We thank the referee for this very stimulating question! We agree that it is unlikely for the phonon energy to transfer into the second laser pulse. We hypothesize that, rather,

the energy is distributed into the phonon bath due to the overall heating of the lattice captured by the model through the two-temperature component. We have modified the manuscript on page 6 starting at line 207:

“...as seen from experiments in Figs. 1 and 2. The phonon bath absorbs the excess energy of the acoustic phonon if the second pulse decreases the oscillation amplitude. The multiple pulses, spaced at...”

R1P5: I think the manuscript falls short in giving a convincing argument beyond plausibility and would greatly benefit of a more elaborated discussion regarding this. In spite of this short comes, I think the data are of good quality, they show a clear example of vibrational coherent control which deserve to be published once the point raised is taken into consideration and properly discussed.

We thank the reviewer for the recommendation; we hope that the changes we introduced to answer their questions and the questions of the reviewer 2 are sufficient.

R1P6: I have also some other minor input which could help the authors in making their manuscript more accessible:

- What do they mean by "second Harmonic periodic lattice distortion"? I find this wording a bit confusing.

We thank the reviewer for pointing this out, we have clarified in the text on page 2 starting at line 56 (see also response to R1P3) that the PLD period is harmonically related to the period of the SDW:

“The periods of the CDW and the PLD are harmonically related to the period of the fundamental SDW: both have double the SDW period.”

Reviewer 2

R2P1: “In the present manuscript Gorobtsov et al discuss an X-ray diffraction experiment on Cr films. The experiment is done in a pump-probe fashion, in which the novelty is a double-pulse excitation prior to the probe pulse, and the expansion of this toward shaping coherent lattice dynamics. While the ideas are interesting and the simulations appear to reproduce experiments, the manuscript lacks significant amounts of even basic information required for reasonably understanding the work. This severely limits the ability to pass reasonable judgement on several claims and conclusions made in the text, and also on the execution of the work itself. The manuscript needs quite

significant alteration before it may rise to a reasonable level, let alone to that of Nature Communications.

Also, a metaphorical elephant in the room is the striking resemblance to an existing work with the same novelty and similar time scales (double-pulse ultrafast X-ray diffraction experiment on a CDW system) in Neugebauer et al, Phys. Rev. B 99, 220302(R) (2019) (This reviewer is not a co-author of that work)”

No persistent oscillation to suppress

Left: Figure from Neugebauer et al, Phys. Rev. B 99, 220302(R) (2019)

Right: Figures from our experiment.

We thank the reviewer for pointing out this work. The paper of Neugebauer et al, Phys. Rev. B 99, 220302(R) (2019) is different in several significant respects. Therefore we firmly believe it does not compromise our novelty:

- 1) They do not observe a persistent oscillation - they only observe one “period”, i.e. a simple excitation - do not enhance it further and do not suppress it (for the obvious reason that there is nothing to suppress or enhance by the time the second pulse arrives, as evident from Fig. 1 in Neugebauer et al we inserted here - compare with our Fig. 1).
- 2) Moreover, the observations in the Neugebauer et al are functionally equivalent to two independent pulses coming after each other, since the excitation is essentially damped by the time a second pulse arrives. There is no control performed in the experiment in Neugebauer et al. Two optical pulses by

themselves do not provide control. Strikingly, we eliminate the phonon after one oscillation (see the top figure of our data above), which appears impossible in the system Neugebauer et. al. studied because of the much stronger damping of the excited phonon.

- 3) For the reasons outlined above (absence of persistent phonon), no differences from the pulses arriving within an oscillation period from each other are observed in Neugebauer et al (compare with Fig. 2 in our work).
- 4) The system in Neugebauer et al is also completely different - it is not a SDW system. Additionally, strong damping of the acoustic phonon in the system prevented the authors from observing **interference**.
- 5) Additionally, we provide a quantitative understanding through the theoretical modelling and use it to propose coherent control protocols with precisely timed multipulse sequences.
- 6) Apart from the weaker phonon damping allowing phonon control and higher quality data allowing quantitative interrogation, our theoretical description is also more substantial than that presented by Neugebauer et al. While Neugebauer et al. consider a single order parameter (the instantaneous PLD), we describe the system with two order parameters: the SDW and the instantaneous PLD. The repeated quench and recovery of the SDW and its interaction with the PLD are absolutely necessary for interpreting our control experiments.
- 7) This discussion goes beyond the scope of our paper; nevertheless, we believe our model would describe the data in Neugebauer et al. with a higher fidelity.
 - a) Our model would apply to their data with a change of the interaction. Having a CDW rather SDW order parameter (OP), in light of our theory, the change would be that the coupling $gL^2 y$ would be $gL y$ with L the CDW order parameter (OP).
 - b) Using our model and understanding, we expect that in reality the cooling rate (restoration rate) of the CDW is such that after the initial shift of the potential for the PLD, the movement of the minimum of the PLD parabola is of the same speed as the PLD itself, catching it and completely diminishing the oscillation. It's not caused by unphysical damping.
 - c) In Neugebauer, to combat the issue with high oscillation period for the PLD caused by changing the changing 2nd order term of the potential, they fix some kind of cutoff without much explanation. What's going on in reality is that the oscillation frequency of the CDW indeed changes, but the 2nd order potential of the PLD remains fixed, which is reflected in the oscillations present in their data.
 - d) We argue thus that the CDW is partially melted after the initial first pulse, the PLD potential restores at the same rate as the PLD OP moves inside it, suppressing the oscillation, which then explains why the second pulse

basically causes the same situation a second time when it occurs early enough.

- e) Two remaining things are present in their data that are not trivially explained using our model: a seemingly downward slope of the oscillation equilibrium, and why the second pulse doesn't do anything when it hits too late. We argue that this is due to a slow residual heating of the CDW from the thermalization of the other excited electronic states by the initial pulse, eventually melting it completely. When the second pulse hits before that happens, it still has effect, if it hits after the CDW is fully melted by these residual electrons, nothing happens. The reason why a similar process is not observed in our experiments is that we have a manifestly metallic system, with almost instantaneous thermalization of all the excited electrons with the SDW. In the case of KMoO_3 , the material is a lot less metallic, especially when the CDW is partially restored, so these processes take more time.
- 8) Additionally, the interpretation in Neugebauer et al. seems to be incomplete:
 - a) In their papers they attempt to describe the behavior of the CDW OP and slave OP describing the PLD with a single OP and energy potential. This can only be done in the static case but is invalid in the dynamic case, especially when the two order parameters don't have the same timescale for their dynamics, and their energetics are influenced differently (only CDW is heated).
 - b) It is not true that the potential for the PLD is a double well: it's a slave OP. Since phonons are not heated directly it's inherent potential doesn't change, only the energy term gyL from the coupling to the CDW changes. The coupling term is missing in the description by Neugebauer et al.
 - c) The oscillation of the PLD should thus not change period (i.e. in ωy^2 the oscillator frequency ω does not change, since they are not heating the PLD directly). This is actually seen in their data see fig below, if one traces the minima of the low fluency oscillation data, they coincide with those at higher fluence.
 - d) The attempt to describe the behavior of the two OPs with their simplified model fails to describe the experimental observations and fails to describe the very complex behavior that occurs due to the interaction between OP's with different dynamical timescales.
 - e) To remedy this, the authors add ad hoc ingredients to produce a better fit, still failing to describe accurately the data (for example oscillations are still present after the peak).
 - f) In an earlier paper (ref 14 of Neugebauer) the authors insert a time dependent damping, claiming that this makes sense since high excitation

might eliminate some scattering channels. This is absolutely not true for the PLD OP, it's the same phonon mode with the same energetics and scattering channels, exciting just means higher amplitude. Furthermore, if that were the case one would expect the damping to also be dependent on the fluence, which is not the case, and from their fit it seems that damping is greater at higher fluence not the opposite.

In summary, in Neugebauer et al, the physics are principally different; there was no interference/coherent control demonstrated in Neugebauer et al (largely due to the system choice), and our theoretical description considers a combination of order parameters, and we fit the evolution path through the free energy – not just an oscillator fit.

We thank the referee for raising this concern and clarify on page 2 starting line 66:

“...transiently enhancing the PLD above its value in equilibrium [5]. Additionally, coherent control requires long coherence time of the system, and we specifically exploit long lifetime of the excited acoustic phonon in chromium compared to other materials (e. g. [24]). The ultrafast recovery of the SDW...”

R2P2: “This is a diffraction experiment, and two PLD reflections (“PLD fringes” in Fig. 1a) are supposedly probed. However, there is no explanation of how the authors got to their observable “PLD amplitude” from the intensities they observe. Furthermore, the

only raw data shown are series of fringes in 3 insets of Fig 1 which are not clearly labelled.

Which of these spots is used? “

We thank the reviewer for pointing out our excessive brevity on this subject. In diffraction from a thin film, the PLD amplitude is linearly proportional to the change in the Laue fringe intensity, as has been established previously in Refs. [8],[20],[24](Singer et. al. 2015, 2016). We have both added a Supplementary Materials, Chapters “The geometric configuration of the X-ray diffraction peak” and “Data collection and analysis”, modified Fig. 1 to show which fringe is chosen, and also modified in the main text on page 2 starting at line 75:

“...making them a powerful tool to study PLD dynamics [7]. In thin films, boundary conditions enforce half-integer ($N+1/2$) number of PLD periods [17] [25] [21], corresponding a PLD wave vector to a certain Laue fringe. The X-ray scattering intensity on the Laue fringe is directly proportional to the magnitude of the PLD [25] [21] [8] (see also Supplementary material), enabling a quantitative measurement of both amplitude and phase of the lattice oscillation....”

R2P3: “How is it used to reach the PLD amplitude shown in the other figures? Is there a structure factor calculation behind this? If not, what is done and how is this justified? Such steps should be clearly spelled out, especially in the methods, if this work is to be believed and its quality assessed.”

We thank the referee for pointing out the missing information for understanding the experimental procedure and analysis. Though the analysis is identical to ref [8], measured at the same instrument on the exact same sample, for clarity we append a supplement to the paper describing the details.

In continuation of the previous point, the results rely on linear relationship between the PLD amplitude and the scattering intensity change in the fringe due to interference, which has been established in [8],[20],[24]. We added a Supplementary on this (see Supplementary, Eq. 4 and the discussion after), which elucidates how this relationship is obtained from a structure factor perspective. We have also added Supplementary Materials, Chapter “Data Collection and Analysis” on how we get to the (normalized) PLD amplitude – in short, the intensity without PLD is subtracted and the result is normalized on the amplitude before the excitation.

R2P4: “2. There are no error bars associated with any of the data. With 100 pulses per delay and 120Hz, the statistics behind these data should be clearly addressed. Without this how would a reader assess/believe the confidence in such data?”

We thank the referee for noticing this. We included the error bars in the new version of the figures and describe their calculation in the figure captions and Supplementary Materials (Chapter “Data collection and Analysis”).

“...Solid lines are experimental data (circles with vertical bars showing uncertainties, estimated as a standard deviation at $t < 0$) connected....”

R2P5: “3. Fluences – how are the Fluences calculated? Is the similar spots of the pump and the probe taken into account? Are these absorbed or incident fluences? If these are absorbed fluences, how does the reflection geometry affect this calculation? What is the penetration of the pump and how was this calculated? Does this account for reflectivity? How?”

The fluences in the paper are the incident fluences measured with a photodiode. As is standard in this type of experiments, the probe size is an order of magnitude smaller than the pump size to have a uniform intensity distribution. The optical skin depth in Cr at 800 nm wavelength is approximately 30 nm, as previously reported [Rakic 1998, Singer 2015], meaning that the film is fully penetrated. We have also previously conducted a thorough investigation on the same setup with the same sample in ref. [8], where a thorough investigation was conducted by using the Bragg peak position and the thermal expansion coefficient as a thermometer. We add in the main text and in the figure 1 caption:

On page 3 starting at line 89:

“...The first 40 fs laser pulse quenches the electronic and spin order [29] throughout the entire 28 nm Cr film as the optical skin depth in Cr at 800 nm wavelength is approximately 30 nm [30] [25].”

On page 3 starting at line 66:

“...that when the incident energy flux at the film...”

In a Fig. 1 caption:

“...The dashed red line marks the time of second pulse arrival. The incident pulse fluences are $P_1 = P_2 = 1.5 \text{ mJ/cm}^2$”

We also clarified in Methods:

“... The sample was excited by optical (800 nm, 40-fs), p-polarized laser pulses propagating nearly collinear with the x-ray pulses. The incident fluence was measured with a photodiode and verified by fluence-dependent measurements to be consistent with the previous measurements in [7]. The final temporal...”

R2P6: “Why do you write “>” on line 118?”

We thank the referee for pointing out the unclear language and clarify on page 4 starting at line 128:

“...the short-term electron temperature below T_N . Another important observation is the saturation effect: at a fluence of around 10 mJ/cm² or higher, the overall temperature...”

R2P7: “4. Several, even basic, experimental details are missing

a) At what temperature was this experiment done?”

We added to the Methods:

.. with a repetition rate of 120 Hz. The film was cooled down to 115 k with a cryojet. Due to the mosaic spread...

R2P8: “b) What X-ray polarization was used? Since you allude to a diamond polarization analyzer, what polarization channel was selected? (I presume sigma-sigma)”

We added to the Methods:

“...diamond crystal monochromator. The x-ray polarization is horizontal, and the scattering geometry is vertical. X-ray diffraction...”

R2P9: “c) There is little account of the experimental setup. Was this in reflection geometry? (assumed from the (200) spectral reflection used)”.

We clarified the experimental geometry with a new supplementary figure (Fig. S3) and accompanying section “The geometric configuration of the X-ray diffraction peak”. The experiment was indeed performed in Bragg (reflection) geometry. We also modify in the Methods:

“... X-ray diffraction in the vicinity of the specular out of plane (002) Bragg peak...”

R2P10: “d) Sample – how do you know TN is lower than bulk? Where did the number 290K come from?”

It has been reported earlier that the critical temperature in thin Cr films is lower than in the bulk due to dimensional crossover (see reference [2] in Supplementary (Zabel 1999) and for experimental measurements references [16], [17] in the main text (Kummamuru and Soh, 2008 and Soh and Kummamuru, 2011). The 290 ± 5 K temperature for a 30 nm thin film was estimated from the references [16], [17], reporting similarly grown films on identical substrates. We thank the reviewer for pointing out the missing references. We modified the text to include them on page 2 starting at line 58:

“...The high Néel temperatures (311 K in bulk [2], decreased due to dimensional crossover [15] to 290 ± 5 K in the 28-nm film studied here [16] [17]) render the critical behaviour close to the SDW phase transition highly accessible”

R2P11: “5. The authors discuss the temperature of the system and subsystems with respect to TN, eg. in line 118. How is this known? Are Debeye-waller dynamics collected and analyzed for assessing lattice temperature?”

For the system temperature, we have the initial temperature (115 K) and the temperature after thermalization (from the PLD value after the phonon damping, ~ 250 K). We have added in the Supplementary material a section on “Thermal effects and temperature estimation”.

The ratios between the heat capacities of the phonon bath and the electronic system can be estimated experimentally through ARPES as ~ 5 (Nicholson et. al., 2016), consistent with what we obtain for dynamical process in TTM.

R2P12: “6. There is some basic confusion surrounding the justification of the simulations. The entire experiment deals with a lattice which is not in a thermal state because of the excited phonons (it cannot be described by Bose-Einstein statistics). However, the simulations deal with temperatures. How is this justified? This should be addressed in detail.”

We describe the PLD by the amplitude $y(t)$, not temperature. The Hamiltonian of the full system could be taken in the form:

$$H=H_{\text{PLD}}+H_{\text{el}}+H_{\text{latt}}+V_{\text{SDW-PLD}}+V_{\text{SDW-el}}+V_{\text{el-latt}}+\dots$$

The electronic system H_{el} consisting of SDW and all other electronic degrees of freedom thermalizes on a 100 fs timescale and can therefore be characterized by the temperature (see for ex. [29] (Brorson et. al., 1990), [35] (Hostetler et. al., 1999)) for the purposes of describing the ps timescale dynamics in our experiments. H_{PLD} describes the PLD as a harmonic oscillator with a damping. Experimental results point to low

damping, suggesting weak coupling to the other phonons. The bath is solely introduced to describe cooling of electrons and is assumed to consist of other phonons, weakly coupled to the system and anharmonically coupled to each other so they thermalize. This assumption is not always correct, and there are examples of memory effects from the bath. However, this assumption is justified in this particular case by the agreement of the simulated dynamics with the experiment.

We thank the referee, and we significantly expanded the model description in the main text to address the details (lines 137 – 173 in the revised text):

“Now we rationalize the results with a non-equilibrium phenomenological model based on a combination of Landau-type theory and two temperature model. The laser pulse with power $Q_{ph}(t)$ acts to increase the energy of the electronic subsystem, which thermalizes on 100 fs timescale, and is therefore thermal on the ps timescales of our interest. The SDW is a part of the electronic subsystem and is thus characterized by the same temperature T_L . On a longer timescale the electronic subsystem cools by transferring the energy to the lattice phonon bath (excluding the PLD mode) with temperature T_b , which thermalizes through anharmonic phonon interactions. The temperature evolution of the electronic system absorbing photons and exchanging heat with the bath is described by [29] [35]

$$C_L \dot{T}_L = -k[T_L(t) - T_b(t)] + Q_{ph}(t), \#(1)$$

$$C_b \dot{T}_b = -k[T_b(t) - T_L(t)]. \#(2)$$

Here C_L, C_b are the heat capacities of the electronic subsystem and the bath, k is the thermal coupling, and $Q_{ph}(t) = \frac{A}{\xi\sqrt{2\pi}} e^{-\frac{1(t-t_0)^2}{\xi^2}}$ is the heat injected by the photon pulses, ξ is the pulse time constant, and A the pulse strength.

The changes in the T_L induced by the laser pulses affect the SDW amplitude as described by Landau-type theory [36] with order parameters L and y denoting the amplitudes of the SDW and the PLD, related to the Fourier component of the spin density at the SDW wave vector q , $L = S_q$ and to the acoustic phonon amplitude $y = u_{2q}$, respectively (Fig. 3a). The essential energetics of the interacting PLD and SDW can be captured by the Landau free energy,

$$F(L, y) = \frac{\alpha}{2}(T_L - T_N)L^2 + \frac{\beta}{4}L^4 - gL^2y + \frac{\rho y_0^2 \omega_0^2}{2}y^2 + \frac{b}{4}y^4, \#(3)$$

where T_L is the temperature of the spin subsystem, T_N is the Néel temperature; the terms with α and β describe the double-well potential for the SDW amplitude L , and the phonon with amplitude y is characterized by the density ρ , displacement amplitude y_0 , frequency ω_0 and anharmonicity b . The lowest-order interaction term between SDW and

PLD with the coupling constant g (akin to [37] [38]) contains the time reversal-odd L squared, for the energy to be time reversal-even, and describes exchange striction. It leads to the force on the phonon mode $f = -\frac{\partial F}{\partial y} = gL^2$ which drives the acoustic phonon amplitude y , as described by an oscillator equation, that follows from Euler-Lagrange equations with the potential (1),

$$\rho y_0^2 \ddot{y} = gL^2 - \rho y_0^2 \omega_0^2 y - by^3 - \gamma \dot{y}, \quad (4)$$

with damping γ and dots designating time derivatives.”

R2P13: “7. Eq. 1 – what is F a function of? Is T_SDW the same as T in line 151? The parameters are mentioned in line 134, but what are their values? How do they relate to literature? Which of these are fit parameters? In lines 325 and 326 the fit results for several parameters are given, with no context. Their values and meaning should be addressed/justified on physical grounds, otherwise this model might as well just be non-physical. There are not even units assigned.”

We have rewritten and expanded the theoretical section in the main text (see **R2P13**) to address what the free energy is a function of (eq. (3)), and also included in the main text parameter values and their units and expanded on the meaning of the fit parameters and their comparison with literature to make sure that the variables are physically consistent (lines 173-186 in the revised text):

“We fitted the solutions of (1-3) to the measured PLD amplitude $y(t)$ to obtain the model parameters $\alpha = 1.6 \cdot 10^7 \frac{J}{K m^3}$, $\beta = 1.55 \cdot 10^{11} \frac{J}{m^3}$, $g = 1.1 \cdot 10^3 \frac{J}{m^3}$, $\frac{\omega_0}{2\pi} = 2.24$ THz, $b = 1.1 \cdot 10^{12} \frac{J}{m^3}$, $\rho = 7150 \frac{kg}{m^3}$, $y_0 = 0.5 \cdot 10^{-12} m$ [5], $\gamma = 1.4 \cdot 10^{-9} \frac{J \cdot s}{m^3}$, and for parameters in (1), fixing $C_L = 1.4 \cdot 10^4 \frac{J}{m^3 K}$ [6], we obtained $C_b = 7.57 C_L$, $k = 3.74 \cdot 10^{16} \frac{W}{m^3 K}$, $A = 2.862 \cdot 10^6 \frac{J}{m^3}$, $\xi = 40$ fs.

The magnitudes of L and y are dimensionless in the model, however the known displacement amplitude y_0 , the measured frequency ω_0 and the material density ρ fix the energy scale in (2) and (3), while C_L fixes the parameter values in (1). Values for heat capacities obtained from fitting are within the expected range: the ratio between the lattice and electronic heat capacities (~ 7) is comparable to that extracted from previous experiments [31], as is the value of thermal coupling constant k [35]. The fitted laser pulse duration ξ agrees with the known pulse duration, and the value of A suggests that less than 1% of total laser intensity is absorbed. With these fitted parameters, our theoretical model provides an excellent quantitative agreement with the experiment, as Figs. 3 (d, e) demonstrates.”

R2P14: “8. Theoretical model: what is capital Phi? (line 313) It appears very central to the model. Based on the sketch in 3a I assume it relates to the SDW. How does it relate to L?”

We thank the reviewer for noticing our typo that we did not correct from the previous version of the paper. Capital Phi is the same as L, which we now corrected.

R2P15: “9. “g” in the model – what is this parameter based on? Does it relate to exchange striction in the system? Is there previous literature on this? Most likely several decades old.”

Parameter g does represent exchange striction. Since it is coupling between two order parameters, it is more akin to references [37] and [38] in the text (Harter et. al. (PRL 2018) and Harter et. al. (Science 2017)) - than to the older [29] (Brorson et. al. 1990) or others – where the model is essentially pure TTM.

We have changed in the text on page 5 starting at line 165:

“...The lowest-order interaction term between SDW and PLD with the coupling constant g (akin to [37] [38]) contains the time reversal-odd L squared, for the energy to be time reversal-even, and describes exchange striction. It leads to the force on the phonon mode $f = -\frac{\partial F}{\partial y} = gL^2$ which drives the acoustic phonon amplitude y , as described by an oscillator equation, that follows from Euler-Lagrange equations with the potential (1),”

R2P16: “10. The two-temperature model (TTM) – There is significant context missing. There is a clear lack of detail and justification surrounding how TTM relates to Eq. 1. To my understanding, this model is coupling the SDW to part of the lattice. Its purpose is to describe heat injection from the laser.”

Eq. 1 is the lowest-order symmetry allowed interaction between the SDW and the PLD. We thank the reviewer for pointing out this opportunity to clarify our treatment. The electronic system that includes SDW thermalizes on a fast timescale of 100s of fs ([29],[35]), compared to the phonon timescale we are interested in. The pump pulses heat up the electronic system and thus control the equilibrium SDW amplitude obtained by the minimization of Eq. 1 with respect to L. That introduces the time dependence in the force acting on the PLD, proportional to $\sim L^2$. The schematic justification of the two-temperature treatment has also been addressed in **R2P12**.

R2P17: “But it appears to assume that the laser couples to the SDW. Is this true? How is this justified and what are the limits of this assumption?”

The reviewer is correct in pointing out that in general, the SDW mechanism is not the only possible excitation mechanism in Cr. In our experiment, the SDW is the primary order parameter, and the PLD is caused by it. The ultrafast quench and reordering of the SDW after a photoexcitation has been observed by tr-ARPES in Cr (Ref. [31] in the manuscript, (Nicholson et. al., 2016)). In addition, the behavior we observe in the experiment relies specifically on the proximity to the magnetic order critical point. In this paper, we specifically exploit the fast changes of the order parameter near the critical point to produce changes that are significantly stronger than with a conventional displacive mechanism.

We modified the text to make the statements clearer. On page 3 starting at line 89:

“The first 40 fs laser pulse quenches the electronic and spin order [29] throughout the entire 28 nm Cr film as the optical skin depth in Cr at 800 nm wavelength is approximately 30 nm [30] [25]. The PLD is then released as an acoustic phonon with wavevector normal to the surface and an oscillation period of approximately 450 fs. In less than a picosecond, the electronic subsystem thermalizes with the lattice below the Néel temperature [29], and the SDW order recovers [31], inducing the recovery of the PLD [8].”

And on page 3, line 110:

“...An excitation of such a magnitude is reminiscent of impulsive excitation of coherent phonons [33]: the PLD recovers to 80% of the original value long before the acoustic phonon starts to dampen. In the conventional displacive excitation, the total PLD change is about the same as the excitation magnitude [34]. We specifically use the critical behaviour of the order parameter in proximity to the critical point to induce significant changes without fully destroying the SDW.”

R2P18: “What is the SDW coupled to? A bath of all phonons that are not coherent? Why is this justified? Please consider existing works on the non-thermal lattice even in simple metals such as Al and Sb (PRB B 95, 054302 (2017) and PRX 6, 021003 (2016), again: not a co-author), or momentum-resolved works by M. Trigo at LCLS.

The SDW is coupled to PLD by the interaction term with g in Eq. 3. We neglect the effect of the excited acoustic phonon on temperature, and the phonon would not exist in equilibrium (the model is non-thermal). The model itself consists of two parts, free energy (with two order parameters) and the TTM. Since the SDW is part of the

electronic system which thermalizes very quickly, we assign to it T_L , the electron system then thermalizes with the phonon bath with temperature T_b on a longer timescale set by heat transfer rate k and the heat capacities.

As in **R2P12** and **R2P16** we thank the referee for pointing out the lack of details in our theoretical description, which we have now corrected (lines 137 to 212 in the revised text, quoted in **R2P12&R2P16** responses).

R2P19: Lastly, does this relate to the TTM analysis done on Cr in ref 26? Also, how were the heat capacities of the subsystems computed? Are they even physical?

The two-temperature part of the model is indeed related to [Brorson et. al., 1990] or [Hostetler et. al., 1999]. The heat capacities are defined to a constant, however their ratio is close to observed in [Nicholson et. al., 2016]. To allow for direct comparison with references, we have changed the units of the parameters involved with the TTM, and defined the constant so that C_L corresponds to [Hostetler et. al., 1999]. At page 5, starting at line 176, we added:

“...and for parameters in (1), fixing $C_L = 1.4 \cdot 10^4 \frac{J}{m^3K}$ [6], we obtained $C_b = 7.57 C_L$, $k = 3.74 \cdot 10^{16} \frac{W}{m^3K}$, $A = 2.862 \cdot 10^6 \frac{J}{m^3}$, $\xi = 40$ fs.

The magnitudes of L and y are dimensionless in the model, however the known displacement amplitude y_0 , the measured frequency ω_0 and the material density ρ fix the energy scale in (2) and (3), while C_L fixes the parameter values in (1). Values for heat capacities obtained from fitting are within the expected range: the ratio between the lattice and electronic heat capacities (~ 7) is comparable to that extracted from previous experiments [31], as is the value of thermal coupling constant k [35]. The fitted laser pulse duration ξ agrees with the known pulse duration, and the value of A suggests that less than 1% of total laser intensity is absorbed.”

R2P20: 11. The last point in this work is a demonstration of engineering the dynamics based on the model. While these are nice ideas, there is an absolute lack of information on this, just fancy figures. Please present details that explain and justify what is happening here.

We have expanded the discussion of the last point/figure as follows:

“...The excellent agreement between theory and data allows us to speculate on the broader control possibilities based on the theoretical parameters extracted from our experiments. By further splitting the laser pulse into a longer pulse train, a further control over a sustained excited state would be possible (Fig. 4). Simulations performed

for Figs. 4 (b, c) using our experimental model demonstrate a possibility to drive the phonon to follow the desired trajectory, for example with sawtooth (Fig. 4 (b)) and sinusoidal (Fig. 4 (c)) oscillation envelopes. We select a low enough fluence to avoid excessive heating of the material (dynamic equilibrium: all heat brought in can be removed by the cooling system). The limiting factors for reaching higher oscillation amplitudes are finite pump pulses, cooling rate, damping and heat capacity C_b . Furthermore, the starting temperature imposes an upper limit on the maximum force per pulse, achieved when SDW is melted completely, and the oscillation frequency limits the time window wherein the force is effective in increasing the oscillation amplitude. Careful balance between these considerations determines the optimal pulse train for a given envelope. In practice, we fix the pulse fluence and optimize the pulse timings one period at a time, allowing for multiple pulses (15 in this simulation) during one period. Fixed pulse fluence and variable timings mimic experimental capabilities. In Cr, the intensity of an individual pulse in such a train would have to be kept below approximately 0.1 mJ/cm^2 so that the total deposited heat dissipates into the substrate and the equilibrium temperature does not rise above T_N ...

R2P21: 12. The discussion begins (line 176) with a statement that suggests that this work establishes an XFEL with pump pulses as a promising route to control. I do not understand this statement. Do the authors suggest that devices will have small XFELS in them? Surely this statement can be substituted with a more reasonable statement on the relevance of this work.

The reviewer is of course correct, we do not suggest tiny XFELs in devices. We clarified the text on page 6, starting at line 233

“...This work establishes an FEL X-ray probe combined with multiple optical photoexcitations in the vicinity of a critical point as a path to rational design of ultrafast control over photoexcited vibrational states and better physical understanding of underlying processes. The ultrafast X-ray probe serves as a high precision feedback to the optical double pump setup, enhancing possibilities for control.”

R2P22: 13. When discussing details in fig. 2a, b and 3d, it would help to clearly define the cases that are discussed in a point-by-point basis. As these panels are rich in detail and very similar, the current descriptions are hard to follow.

We thank the reviewer for pointing it out. We have reorganized Fig. 2 and have also changed in text:

On page 3 starting at line 117:

“Figure 2 (a) demonstrates a full map of how the amplitude of the PLD changes with both $\tau_2 - \tau_1$ and the probe delay t for 2 laser pulses of the same fluence 1.5 mJ/cm^2 ...”

On page 4 starting at line 125:

“...the overall temperature of the system without displacive excitation [34]. Remarkably, the second pulse can be significantly weaker, but the effect will still remain: Fig. 2 (b) shows the full map when the second pulse is weakened by a factor of 2, leaving the short-term electron temperature below T_N . Another important observation is the saturation effect: at a fluence of around 10 mJ/cm^2 or higher, the overall temperature of the system rises above T_N , and the system is indifferent to the second pulse arrival time τ_2 : compare the “enhanced” and “suppressed” phonons in Fig. 2 (c), which are virtually the same. At the fluences of and higher than 10 mJ/cm^2 , the magnetic order is destroyed...”

And in the figure caption:

“Map of the PLD amplitude before and after excitation by two pulses with fluences $P_1=1.5 \text{ mJ/cm}^2$, $P_2=0.7 \text{ mJ/cm}^2$. c, Magnitude of the PLD in “enhancement” and “suppression” conditions with laser fluences $P_1=9.5 \text{ mJ/cm}^2$ and $P_2=4.8 \text{ mJ/cm}^2$. Solid lines are experimental data (points, vertical bars represent errors) connected. The dashed red lines mark the times of second pulse arrival. Orange $-\tau_2 - \tau_1$ of 1295 fs, Purple $-\tau_2 - \tau_1$ of 1065 fs.”

R2P23: 14. Why are the Laue fringes commensurate with the CDW peak’s q vector?

The Laue fringes are commensurate with the PLD peak’s q vector because of the boundary conditions on the wave: only half-integer numbers of periods are allowed, as has been established before (Soh and Kumamuru, 2011), which connects the film thickness and therefore Laue fringes to the PLD q vector, measured extensively and explained in [8][21][25](Singer et. al. 2015 & 2016). We have added a Supplementary section “X-ray diffraction from a thin film with a periodic lattice distortion” which details how this arises from the structure factor, with an illustration (Fig. S2). We have also made it clearer in the main text (page 2, starting at line 75):

“...making them a powerful tool to study PLD dynamics [7]. In thin films, boundary conditions enforce half-integer ($N+1/2$) number of PLD periods [17] [25] [21], corresponding a PLD wave vector to a certain Laue fringe. The X-ray scattering intensity on the Laue fringe is directly proportional to the magnitude of the PLD [25] [21] [8] (see also Supplementary material), enabling a quantitative measurement of both amplitude and phase of the lattice oscillation....”

R2P24: 15. Lines 119-120 – compare 2c top and bottom – what do we see? What is the relevant difference? This is not clear.

We have rearranged Fig. 2 to make it clearer and have made changes in the text for clarification, listed our answer to point 13. In short, the relevant difference is that the timing of the pulses is different, but the result is the same – because the film temperature after the first pulse is above T_N .

R2P25: 16. Line 120 – magnetic order is destroyed and does not recover – how do you know this?

The second pump pulse induces no significant effect on the phonon (Fig. 2, c). We have reworked Fig. 2 to better illustrate the point and have clarified in the text (page 4, starting at line 128):

“Another important observation is the saturation effect: at a fluence of around 10 mJ/cm² or higher, the overall temperature of the system rises above T_N , and the system is indifferent to the second pulse arrival time τ_2 : compare the "enhanced" and "suppressed" phonons in Fig. 2 (c), which are virtually the same.”

In addition, at the fluence of 10 mJ/cm² it has previously been measured (Singer 2016) that after the phonon is damped and the film is in thermal equilibrium with itself (but not substrate), there is no PLD => above T_N .

R2P26: 17. Lines 142 and 151 – the mathematical relations, how were they derived?

We thank the reviewer for turning our attention to these, we have corrected a typo in the equation for the force (minus) and we have expanded the explanation in the main text at page 5, line 188:

“Below T_N , the non-zero PLD is induced by the SDW order L . Below T_N , PLD magnitude varies as $L \sim \sqrt{T_N - T_L}$ within the mean-field approximation, as can be found by minimizing the free energy (1) with respect to L .”

R2P27: 18. Figures:

a- the figures and panels are not cited in order.

b- Fig 1a – the shape of this colorful potential is not clear, and the colorscale is shown with no units or numerical scale.

The potential surface is purely illustrative, which is why the units or numerical scale are not listed. We have added in the figure description:

“...surfaces (not to scale) show...”

c- Fig 1a inset is not clear. What are the indicated directions and what is q ?

We have reworked the figure and added in the figure description:

“...Here q is the scattering vector..”

d- Fig 1 – there are no labels or units around the images of diffraction peaks. Not even a color scale

We have added the color scale and unit label

e- The lower inset of 1b looks like a convincing piece of this work. I would suggest to show it clearly. Also please indicate the arrival of the 2nd pump in a similar way to the other delay scans.

That is a great suggestion! We have reworked Fig. 1 to bring the inset out as a main figure and marked the 2nd pump arrival time.

R2P28: f- Fig2a,b – differences between these two panels are not clear. A more suitable color scheme or annotation for guidance would be helpful.

We have reorganized the figure, changed the caption accordingly and made an annotation on the figure to help trace it.

R2P29: g- Fig. 3b,c – what are the conditions of this simulation?

Fig. 3 b, c are schematic and the features of the potential surfaces are exaggerated to make them clear. We have added in the caption to clarify:

" Features of the potential surfaces are exaggerated, and the color accentuates the surface shape, for illustration purposes."

R2P30: h- Figure 4b,c - are the black dots supposed to be the red dot?

We corrected the discrepancy.

R2P31: 19. This work is about Chromium, an important material that exhibits an SDW, inducing a PLD, and an associated CDW. This is a relatively unique system, or at least a very well known one. But the name Chromium is not in the title nor the abstract. I would suggest to change this. Also, why is Ref. 5 in a abstract in the context of non-equilibrium?

We thank the reviewer for pointing this out. There was a mix-up in the bibliography list, we corrected it (Ref. 5 in the abstract was supposed to be Zong et. al, 2018). Regarding chromium, we added in the abstract (line 32):

"...near the critical point of the SDW **in chromium**. We apply..."

EDITORIAL REQUESTS

POLICIES AND FORMS REQUIRED FOR RESUBMISSION

We have addressed the referee questions in this point-by-point response. We have uploaded the required checklist together with the revised manuscript.

DATA AND CODE AVAILABILITY

We have added the data availability section:

"Data Availability

Raw data were generated at the Linac Coherent Light Source (LCLS), SLAC National Accelerator Laboratory large-scale facility. Derived data supporting the findings of this study are available from the corresponding author upon reasonable request. Source data used for figures are available in the Open Science Framework repository at https://osf.io/rgf8h/?view_only=c7a8cd7244044b638c748c0b3b8ce106.

"

We did not use bespoke custom mathematical algorithms/computational analysis software for producing our data.

ORCID

Corresponding author has linked his ORCID and informed other authors that they can link their ORCID to nature accounts before acceptance.

REVIEWER COMMENTS

Reviewer #1 (Remarks to the Author):

I recommend the manuscript for publication

Reviewer #2 (Remarks to the Author):

The authors have made many changes to this manuscript. This is a very good direction considering its extremely low quality in the previous round (that is not an opinion about the scientific work, but about the manuscript presenting it). Having more coherent descriptions of the work, including basic information like the units associated with the numbers presented, allows now for a reasonable consideration of the results.

I find the experiment well-planned, and the results interesting. I believe that following some further clarifications, the manuscript may be of sufficient quality for publication.

Specifically, several additions to the supplement have been made. These are good, and in some cases quite necessary information for understanding the main text, suggesting that some key information should migrate to the main text or the methods section.

I thank the authors for the detailed discussion about the Neugebauer et al paper. Indeed the underlying physics of the two systems differ, and the extent of the work and conclusions do too. The ideas and experimental scheme are nonetheless similar, and I think its good that it is at least cited now.

Here are points to consider:

1. Line 110-114 – I find this discussion very confusing. In line 111 is the PLD equated to the acoustic phonon or is this a grammar issue? When does it reach 125% and in which of the two plots?
2. Figures 1a,b – the amplitude reaches negative values. With the changes in this version I can now understand that this is a value extracted from the intensity riding on top of the thickness fringe, but a value associated with the PLD amplitude cannot be negative (assuming that we are still discussing a wave with an amplitude and a phase). The meaning of this PLD amplitude needs to be defined in the text, as it is the main quantity discussed. Earlier in the text (line 83), it is stated that the measurement allows both amplitude and phase to be observed. Is this quantity a combination of the two? If I understand correctly, the quantity referred to here, is the A_PLD in the new supplement. Is this correct? Coherent terminology is required to follow this discussion, please consider using the same terms in the main text and in the supplement (especially since the supplement is really needed

to understand the analysis). If I understood correctly, then some simplified form of Eq. 4 in the supplement should be presented in the main text (e.g. after approximations) such that a reader could believe or at least understand what the numbers shown in the delay scans actually represent, without having to read the detailed derivation in the supplement. If this still justifies having negative numbers for the PLD amplitude, this must be clearly explained in the main text.

3. Line 118-119 – do I understand correctly that proximity to the critical point means that you chose a specific fluence to have this proximity? Please indicate this.

4. Line 132 – how do you know that the electrons remain below T_N ? No temperature estimates have been discussed at this point. Is this an assumption? The same question for line 134... Is this related to the temperature discussion in the supplement? Please clarify the source of this statement in the main text.

5. In the two-temperature model (TTM), you assume that the spins and the electrons share the same temperature, essentially implying infinite coupling. How is this justified? Normally energy flow in magnetic systems (regardless if ferromagnetic or antiferromagnetic) is described by 3-temperature models because of a non-infinite coupling (see for example the M3TM model in B. Koopmans, Nature Materials 2011). A minor additional note is that “L” is usually used for “lattice”, not for the electrons (in which case “e” is used...). This is just convention.

6. Line 159 – it is written that $\gamma = U_{2q}$. why do we have γ then? It appears later as a dimensionless quantity. Should it be U_{2q} normalized to something?

7. Line 186 – it is stated that less than 1% of the laser is absorbed. Does this agree with the expected reflectivity and transmission of Cr at this angle? This comparison should be stated if absorption is to be discussed.

8. Fig. 1b – in the raw data strip image it appears that (002) is much weaker than the PLD reflection. This is at odds with Ref. [8] and also intuitively unlikely. Is this an error? Is this just a detector snapshot that does not account for the curvature of Ewald’s sphere? Please explain in the text or caption, especially if this is a detector snapshot, as a reader unfamiliar with expert experimental details will not understand the image.

9. Line 351 – The authors state that the error bars are “estimated as a standard deviation at $t < 0$ ”. Please explain this. Standard deviation of what? Also, why is this chosen, and why is this a reasonable error estimate? If this is a standard deviation of a signal before excitation, it surely does not reflect experimental certainty after excitation. Especially when the signal weakens... Also, does this mean that all data points have the same error bars? Please clarify, or choose more reasonable error estimates (see also next comment).

10. In the supplement the data collection section now properly describes that a sum over a ROI produces the intensity collected. Numbers associated with this ROI should be used as reliable errors (e.g. measures of the change in the STD in the ROI, or similar)

Reviewer #4 (Remarks to the Author):

The paper by Gorobtsov et al. reports experimental and theoretical investigation for the coherent control of coherent acoustic phonons in a prototypical spin-density-wave (SDW) system, thin Cr films, using double pump-pulse excitation and X-ray scattering probe. The data exhibited both destructive and constructive excitation of coherent longitudinal acoustic phonon at 2.21 THz.

Although the Reviewer 2 pointed out many technical problems, I have rather fundamental questions arising from the experiments and analysis in the present paper.

1. To tell the truth, I was not surprised with the main data in Figs. 1-2 and simulation results in Fig. 3, since similar information of coherent control of coherent phonons has already been obtained by several groups. For example, Science 247, 1317 (1990), Phys. Rev. Lett. 102, 037402 (2009), Nat. Commun. 3, 721 (2012), Phys. Rev. B 87, 075307 (2013), Nat. Commun. 6, 8367 (2015). That means the present data, simulation, and discussion are mostly explained if we consider only “coherent control of coherent phonons”. The one interesting effect might be the 2 times weaker excited phonon observed when the arrival of the second pump pulse was less than ~ 0.3 ps (Fig. 2a and 2b). However, this could be explained by destructive interference of coherent phonons and/or the saturation effect previously observed at $\sim 3-4$ mJ/cm² single pump in Ref. [8].

2. In the response to the 4th comment from the Reviewer 1, the authors respond that “The phonon bath absorbs the excess energy of the acoustic phonon if the second pulse decreases the oscillation amplitude”. I am not convinced with this statement. The destruction of the acoustic phonon was observed just after the second pump pulse arrived (Fig. 1c), suggesting ultrafast energy dissipation to the phonon bath, if the authors are correct. I am not sure if such the thermal energy transfer occurred in sub-picosecond time scale. It would be useful to apply the three temperature model (3TM) [please refer e.g., Phys. Rev. B. 87, 235124 (2013)] if the authors propose the energy escape into the phonon bath (lower-energy acoustic phonon branches).

3. The authors mention that the frequency of the acoustic phonon remains unchanged in page 3, which would mean the phonon dispersion in the excited state was not perturbed. On the other hand, the Landau free energy and Euler-Lagrange equations have anharmonic terms, suggesting the anharmonic forces will change the phonon dispersion. Please evaluate the strength of the anharmonic term, e.g., “b” in Eq. (4), if this causes any change in the dispersion.

4. Minor point: In the two-temperature model, the electronic heat capacity was treated as a constant, but in reality, C_{L} should depend on the electron temperature, $C_{\text{L}} = \gamma T_{\text{L}}$, where γ is the Sommerfeld constant as in Ref. [29].

In summary, this paper will fall in the minor advance in the experiment after the previous paper by the same group, Ref. [8]. By this and above reasons, I cannot recommend the publication of this manuscript in Nature Communications, but may be publishable in some other physical journals.

Reviewer #1 (Remarks to the Author):

I recommend the manuscript for publication

We thank the reviewer for recommending the publication!

Reviewer #2 (Remarks to the Author):

The authors have made many changes to this manuscript. This is a very good direction considering its extremely low quality in the previous round (that is not an opinion about the scientific work, but about the manuscript presenting it). Having more coherent descriptions of the work, including basic information like the units associated with the numbers presented, allows now for a reasonable consideration of the results.

I find the experiment well-planned, and the results interesting. I believe that following some further clarifications, the manuscript may be of sufficient quality for publication.

We thank the reviewer and have introduced the clarifications, which we list below.

Specifically, several additions to the supplement have been made. These are good, and in some cases quite necessary information for understanding the main text, suggesting that some key information should migrate to the main text or the methods section.

I thank the authors for the detailed discussion about the Neugebauer et al paper. Indeed the underlying physics of the two systems differ, and the extent of the work and conclusions do too. The ideas and experimental scheme are nonetheless similar, and I think its good that it is at least cited now.

Here are points to consider:

1. Line 110-114 – I find this discussion very confusing. In line 111 is the PLD equated to the acoustic phonon or is this a grammar issue? When does it reach 125% and in which of the two plots?

We are grateful to the reviewer for pointing out the unclear phrasing, and we clarified in the text:

“The transient magnitude of the PLD oscillation (half of the difference between the maximum PLD and the minimum PLD within one period) reaches 125% of the unperturbed lattice distortion (see Fig. 1 (d) after the second pump pulse).”

2. Figures 1a,b – the amplitude reaches negative values. With the changes in this version I can now understand that this is a value extracted from the intensity riding on top of the thickness fringe, but a value associated with the PLD amplitude cannot be negative (assuming that we are still discussing a wave with an amplitude and a phase). The meaning of this PLD amplitude needs to be defined in the text, as it is the main quantity discussed. Earlier in the text (line 83), it is stated that the measurement allows both amplitude and phase to be observed. Is this quantity a combination of the two? If I understand correctly, the quantity referred to here, is the A_PLD in the new supplement. Is this correct? Coherent terminology is required to follow this discussion, please consider using the same terms in the main text and in the supplement (especially since the supplement is really needed to understand the

analysis). If I understood correctly, then some simplified form of Eq. 4 in the supplement should be presented in the main text (e.g. after approximations) such that a reader could believe or at least understand what the numbers shown in the delay scans actually represent, without having to read the detailed derivation in the supplement. If this still justifies having negative numbers for the PLD amplitude, this must be clearly explained in the main text.

We thank the reviewer; our terminology may indeed have been unclear. “Negative PLD amplitude” here means literally a negative A_{PLD} if the phase offset ϕ_0 is kept the same ($\phi_0 = const$ due to pinning on the interfaces), i. e. that the sign of the PLD is flipped. We have added some of the information from the supplementary to the text:

“gives rise to a satellite peak on the Laue fringes of the out-of-plane (002) crystal Bragg peak (Fig. 1 (a), inset), where the total peak intensity $I(q, t)$ as a function of a wavevector $\mathbf{q} = (0, 0, q)$ and time after excitation t ([8, 25], Supplementary Materials)

$$I(q, t) = I_0 |F_u(q)|^2 [|f(q)|^2 + q A_{PLD}(t) \sin(\alpha) [f(q)f(q - 2Q) - f(q)f(q + 2Q)]], \quad (1)$$

where I_0 is the normalization constant, $F_u(q)$ is the structure factor of the unit cell, $f(q) = \frac{\sin(Nqa/2)}{\sin(qa/2)}$ describes the Laue fringes, $\alpha = Qa[N - 1] - \phi_0$, PLD is defined as a spatial wave $A_{PLD}(t) \cos(2Qr_n^0 - \phi_0)$, $r_n^0 = n \cdot a$ are the undistorted atomic positions, a is the lattice constant, and $A_{PLD}(t)$ the PLD amplitude, $2Q$ is the wave vector of the PLD, and ϕ_0 defines the offset of the wave. An XFEL pulse 0.2 mm in diameter envelopes multiple SDW domains [32] below the Néel temperature, making the measurement statistical over domains.”

and

“Evolution of the PLD amplitude $A_{PLD}(t)$ as a function of the probe delay t presented in Figs. 1 (c, d) for 2 different $\tau_2 - \tau_1$ shows that we are indeed able to completely suppress or further enhance the excited state with the second pulse. The phase offset ϕ_0 is constant and $A_{PLD}(t)$ is negative when the sign of $A_{PLD}(t) \cos(2Qr_n^0 - \phi_0)$ is reversed.”

3. Line 118-119 – do I understand correctly that proximity to the critical point means that you chose a specific fluence to have this proximity? Please indicate this.

Yes, we chose the fluence specifically to reach the maximum enhancement of the transient PLD amplitude. We clarified in the text:

“Figure 2 (a) demonstrates a full map of how the amplitude of the PLD changes with both $\tau_2 - \tau_1$ and the probe delay t for 2 laser pulses of the same fluence $P = 1.5 \text{ mJ/cm}^2$. The total laser fluence was chosen to reach the maximum enhancement of the transient PLD after one pulse laser excitation.”

4. Line 132 – how do you know that the electrons remain below TN? No temperature estimates have been discussed at this point. Is this an assumption? The same question for line 134... Is this related to the temperature discussion in the supplement? Please clarify the source of this statement in the main text.

The reviewer is correct to point this out, the TN statement comes from the model. We moved the statement in the text to the discussion:

“Since the SDW cools rapidly it is thus beneficial to heat the SDW a bit above T_N , just so the PLD mode can catch up and pass all the way to the minimum at zero (the condition achieved, according to the model, in Fig. 2 (a), but not Fig. 2 (b))”

5. In the two-temperature model (TTM), you assume that the spins and the electrons share the same temperature, essentially implying infinite coupling. How is this justified? Normally energy flow in magnetic systems (regardless if ferromagnetic or antiferromagnetic) is described by 3-temperature models because of a non-infinite coupling (see for example the M3TM model in B. Koopmans, Nature Materials 2011). A minor additional note is that “L” is usually used for “lattice”, not for the electrons (in which case “e” is used...). This is just convention.

We thank the reviewer for the suggestion. Since Cr is a metal and the SDW is created because of Fermi surface nesting, inherently coupling the electronic and magnetic subsystems, it allows us to assign the same temperature to both. This strong coupling is confirmed through the time-resolved ARPES and optical reflectivity in Cr [Nicholson et al., Brorson et. al.]. Additionally, the combination of Landau theory and a 2TM in our simulations reproduces the experimental observations almost perfectly, indicating that 2TM is sufficient.

The choice of letter L was motivated by the routine use of L to denote antiferromagnetic order parameter amplitude. We agree that “L” is also often used for “lattice”, but it is a question of convention choice here.

6. Line 159 – it is written that $y = U_{2q}$. why do we have y then? It appears later as a dimensionless quantity. Should it be U_{2q} normalized to something?

The reviewer is correct, we did in fact normalize y to 1 by means of y_0 , we corrected this in the text:

“...the normalized acoustic phonon amplitude $y = u_{2q}/y_0$, respectively (Fig. 3a).”

7. Line 186 – it is stated that less than 1% of the laser is absorbed. Does this agree with the expected reflectivity and transmission of Cr at this angle? This comparison should be stated if absorption is to be discussed.

We thank the reviewer for pointing this out. We double-checked the parameters in the 2TM part of the model, which are defined from a fit down to a constant multiplier. The constant must be extracted from external sources since it does not directly affect the behaviour of the density waves. We previously used a constant electronic heat capacity C_L estimate from Ref. [J. L. Hostetler et. al., 1999] to extract the multiplier. However, considering that a better approximation is temperature-dependent $C_L = c_L T_L$ (as pointed out by Reviewer #3), using the lattice heat capacity C_b is preferable. We have found parameter values for C_b and k in Cr in [M. Saghebfar et. al., 2016], and we have found the approximate complex refraction index data for thin Cr films in [V. Lozanova et. al., 2014]. After the changes in the heat capacity following [M. Saghebfar et. al., 2016], we found the absorption coefficient of ~15%, which is close to the expected value from the transmissivity and reflectivity measurements in [V. Lozanova et. al., 2014] (~20%). We have modified and expanded discussion in the text accordingly:

“...with damping γ and dots designating time derivatives. We fitted (2-4) to the measured normalized PLD amplitude $y(t) = A_{PLD}(t)/A_{PLD}(0)$, taking $y_0 = 0.5 \cdot 10^{-12} m$ [21] and $\rho = 7150 \frac{kg}{m^3}$ and $L(t = 0) = 1$, to obtain the model parameters in (4) as $\alpha = 5.1 \cdot 10^7 \frac{J}{K m^3}$, $\beta = 9.0 \cdot 10^9 \frac{J}{m^3}$, $g =$

$3.46 \cdot 10^5 \frac{J}{m^3}$, $\frac{\omega_0}{2\pi} = 2.18 \text{ THz}$, $b = 1.16 \cdot 10^4 \frac{J}{m^3}$, $\gamma = 1.37 \cdot 10^{-9} \frac{J \cdot s}{m^3}$. Parameters in (2) and (3) are defined to a constant multiplier (except for the pulse duration $\xi = 40 \text{ fs}$) by fitting. The constant can be fixed by estimating based on literature $C_b = 3.23 \cdot 10^6 \frac{J}{m^3 K}$ [37], obtaining $k = 1.25 \cdot 10^{18} \frac{W}{m^3 K}$, $c_L = 1.91 \cdot 10^3 \frac{J}{m^3 K^2}$, mean value of $A = 8 \cdot 10^7 \frac{J}{m^3}$.

The magnitudes of L and γ are dimensionless in the model, however the measured displacement amplitude y_0 [21], the measured frequency ω_0 and the material density ρ fix the energy scale in (4) and (5), while the bath/lattice heat capacity C_b of Cr [37] fixes the parameter values in (2) and (3). The anharmonic term gives a small variation in frequency between excitations ($b/\rho y_0^2 \omega_0^2 \sim 0.03$). Values for heat capacities obtained from fitting are within the expected range: the ratio between the lattice and electronic heat capacities during the excitation ($C_b/c_L T \sim 7$) is comparable to that extracted from previous experiments [31] (larger c_L values than in bulk [37] can be explained by disorder in the film [41]), as is the value of thermal coupling constant k [37]. Absorption at 800 nm wavelength and 30° angle can be estimated based on measured complex refractive index $n \approx 3 + 1.1i$ for Cr thin films [42] as $\sim 20\%$, while the obtained value of A gives absorption of $Ad/P \sim 15\%$, which is remarkably close considering possible differences due to film roughness. With these fitted parameters, our theoretical model provides an excellent quantitative agreement with the experiment, as Figs. 3 (d, e) demonstrate.

8. Fig. 1b – in the raw data strip image it appears that (002) is much weaker than the PLD reflection. This is at odds with Ref. [8] and also intuitively unlikely. Is this an error? Is this just a detector snapshot that does not account for the curvature of Ewald's sphere? Please explain in the text or caption, especially if this is a detector snapshot, as a reader unfamiliar with expert experimental details will not understand the image.

We thank the reviewer; this is indeed a detector snapshot. We added to the caption:

“Detector image of the X-ray scattering from periodic atomic displacement on a fringe of the main peak from the crystalline film. Here $\mathbf{q} = 4\pi/\lambda \cdot \sin(\theta)$ is the scattering vector with λ being the x-ray wavelength. Central peak is weaker than the fringes due to the position of the Ewald sphere.”

9. Line 351 – The authors state that the error bars are “estimated as a standard deviation at $t < 0$ ”. Please explain this. Standard deviation of what? Also, why is this chosen, and why is this a reasonable error estimate? If this is a standard deviation of a signal before excitation, it surely does not reflect experimental certainty after excitation. Especially when the signal weakens... Also, does this mean that all data points have the same error bars? Please clarify, or choose more reasonable error estimates (see also next comment).

The reviewer rises a good point. While the relative error increases when the signal decreases, the absolute error does not (since the signal is directly proportional to y). We agree however that the behaviour after the excitation can increase the uncertainty. To account for possible changes in the experimental uncertainty after the excitation, we have added a systematic error contribution defined as a difference between different measurements at the lowest point immediately after the excitation

(which is an upper bound on the error on most of the time interval). We updated the figures and modified in the caption of Fig. 1:

“...control cases. Solid lines connecting circles are experimental data. Vertical bars show uncertainties, estimated as a standard deviation σ_{FEL} of the PLD amplitude at $t < 0$ and as $\sigma_{\text{full}} = \sqrt{\sigma_{\text{FEL}}^2 + \sigma_{\text{ex}}^2}$ at $t > 0$, where σ_{ex} is a standard deviation of the minimum reached PLD amplitude after the first excitation between different scans with identical fluence P . The vertical...”

10. In the supplement the data collection section now properly describes that a sum over a ROI produces the intensity collected. Numbers associated with this ROI should be used as reliable errors (e.g. measures of the change in the STD in the ROI, or similar)

We thank the referee for the proposal, the STD over ROI would overestimate the uncertainty due to the peak shape. We believe that our addition of an error estimate from differences between repeated measurements solves the error question.

Reviewer #4 (Remarks to the Author):

The paper by Gorobtsov et al. reports experimental and theoretical investigation for the coherent control of coherent acoustic phonons in a prototypical spin-density-wave (SDW) system, thin Cr films, using double pump-pulse excitation and X-ray scattering probe. The data exhibited both destructive and constructive excitation of coherent longitudinal acoustic phonon at 2.21 THz.

Although the Reviewer 2 pointed out many technical problems, I have rather fundamental questions arising from the experiments and analysis in the present paper.

1. To tell the truth, I was not surprised with the main data in Figs. 1-2 and simulation results in Fig. 3, since similar information of coherent control of coherent phonons has already been obtained by several groups. For example, Science 247, 1317 (1990), Phys. Rev. Lett. 102, 037402 (2009), Nat. Commun. 3, 721 (2012), Phys. Rev. B 87, 075307 (2013), Nat. Commun. 6, 8367 (2015). That means the present data, simulation, and discussion are mostly explained if we consider only “coherent control of coherent phonons”. The one interesting effect might be the 2 times weaker excited phonon observed when the arrival of the second pump pulse was less than ~ 0.3 ps (Fig. 2a and 2b). However, this could be explained by destructive interference of coherent phonons and/or the saturation effect previously observed at ~ 3 -4 mJ/cm² single pump in Ref. [8].

We thank the reviewer for the opportunity to talk about the differences from the previous work. Coherent control over collective modes has attracted considerable interest in the scientific community over the last decades, as shown by the list of high impact publications provided by the reviewer (there are also more recent examples such as a Horstmann et. al., Nature, 2020 (<https://doi.org/10.1038/s41586-020-2440-4>), published after our submission). Our study represents a novel control mechanism in near-critical conditions using direct interrogation by ultrafast x-rays. Here we control one of the coupled order parameters by exciting another, near its critical point. Partial melting of that order is responsible for the diminished response at low delay. The papers the reviewer

listed are over a wide range of the field, involve different physics than ours, and offer different conclusions; in many of the papers the only commonality is the use of a pump-pump-probe scheme. We disagree with the referee that “present data, simulation, and discussion are mostly explained if we consider only “coherent control of coherent phonons”. We do not drive the mode directly. The reviewer correctly noticed the interesting effect at low pump-pump delay. This feature is the result of partial melting of the SDW that is characteristic of the demonstrated control mechanism and only observable in a double pump measurement and near criticality. For a direct control, at short delay constructive interference would occur in stark contrast to our measurement. To clarify this important distinction in particular, we added to the text:

“We specifically use the critical behaviour of the order parameter (SDW) in proximity to the critical point to exert control of another coupled mode (PLD). This is in contrast to directly driving the target mode [35].”

In the interest of completeness, we also included some of the references provided by the reviewer where relevant:

“...would be possible (Fig. 4). Optical control of molecular motion with long pulse trains has been demonstrated experimentally [43]. Simulations...”

2. In the response to the 4th comment from the Reviewer 1, the authors respond that “The phonon bath absorbs the excess energy of the acoustic phonon if the second pulse decreases the oscillation amplitude”. I am not convinced with this statement. The destruction of the acoustic phonon was observed just after the second pump pulse arrived (Fig. 1c), suggesting ultrafast energy dissipation to the phonon bath, if the authors are correct. I am not sure if such the thermal energy transfer occurred in sub-picosecond time scale. It would be useful to apply the three temperature model (3TM) [please refer e.g., Phys. Rev. B. 87, 235124 (2013)] if the authors propose the energy escape into the phonon bath (lower-energy acoustic phonon branches).

The energy of the acoustic phonon does not directly dissipate into the phonon bath, it goes to the SDW, and then dissipates into the bath. In a swing analogy, this is like moving the swing axis forcefully when the swing is at a maximum height and removing the swing potential energy by doing that work. The energy of the acoustic phonon passed in this way is negligible in comparison to the total heat transfer to the phonon bath after the photoexcitation and does not affect the model in a significant way. We clarified in the paper:

“...in Figs. 1 and 2. If the second pulse decreases the oscillation amplitude, the excess energy of the phonon transfers to the SDW through coupling between SDW and PLD and then dissipates into the phonon bath. The excess energy of the phonon is negligible compared with the energy influx from the laser excitation (experimentally, the long-time damping of the phonon does not lead to a change in the average PLD value [8]). Multiple pulses...”

Regarding 3TM, since Cr is a metal and the SDW is created because of Fermi surface nesting, inherently coupling the electronic and magnetic subsystems, it allows us to assign the same temperature to both. This strong coupling is confirmed through the time-resolved ARPES and optical reflectivity in Cr [Nicholson et al., ref 31, Brorson et al., ref 29]. Additionally, the combination of Landau theory and a

2TM in our simulations reproduces the experimental observations almost perfectly, indicating that 2TM is sufficient.

3. The authors mention that the frequency of the acoustic phonon remains unchanged in page 3, which would mean the phonon dispersion in the excited state was not perturbed. On the other hand, the Landau free energy and Euler-Lagrange equations have anharmonic terms, suggesting the anharmonic forces will change the phonon dispersion. Please evaluate the strength of the anharmonic term, e.g., “b” in Eq. (4), if this causes any change in the dispersion.

We thank the reviewer for the comment and the suggestion, we have corrected in the text:

“The frequency of the acoustic phonon remains largely unchanged...”

and added:

“...fixes the parameter values in (2) and (3). The anharmonic term gives a small variation in frequency between excitations ($b/\rho y_0^2 \omega_0^2 \sim 0.03$). Values...”

4. Minor point: In the two-temperature model, the electronic heat capacity was treated as a constant, but in reality, C_L should depend on the electron temperature, $C_L = \gamma T_L$, where γ is the Sommerfeld constant as in Ref. [29].

We thank the referee for a suggestion, we have introduced the temperature dependence into $C_L = c_L T_L$. This did not change the fit quality or the conclusions, but we did normalize the parameter values in the TT part of the model accordingly based on [Saghebfar et. al., 2016]. We modified in the text to demonstrate these changes:

“Here $C_L = c_L T_L$, C_b are the heat capacities of the electronic subsystem and the bath, c_L is proportional to the electronic Sommerfeld coefficient, ...”

And

““We fitted (2-4) to the measured normalized PLD amplitude $y(t) = A_{PLD}(t)/A_{PLD}(0)$, taking $y_0 = 0.5 \cdot 10^{-12} m$ [21] and $\rho = 7150 \frac{kg}{m^3}$ and $L(t=0) = 1$, to obtain the model parameters in (4) as $\alpha = 4.15 \cdot 10^{13} \frac{J}{K m^3}$, $\beta = 7.42 \cdot 10^{15} \frac{J}{m^3}$, $g = 3.46 \cdot 10^5 \frac{J}{m^3}$, $\frac{\omega_0}{2\pi} = 2.18$ THz, $b = 10.4 \cdot 10^3 \frac{J}{m^3}$, $\gamma = 1.37 \cdot 10^3 \frac{J \cdot s}{m^3}$. Parameters in (2) and (3) are defined to a constant multiplier (except for the pulse duration $\xi = 40$ fs) by fitting. The constant can be fixed by estimating based on literature $C_b = 3.23 \cdot 10^6 \frac{J}{m^3 K}$ [36], obtaining $k = 1.23 \cdot 10^{18} \frac{W}{m^3 K}$, $c_L = 2.11 \cdot 10^3 \frac{J}{m^3 K^2}$, $A = 8.25 \cdot 10^7 \frac{J}{m^3}$.”

The magnitudes of L and γ are dimensionless in the model, however the measured displacement amplitude y_0 [21], the measured frequency ω_0 and the material density ρ fix the energy scale in (4) and (5), while the bath/lattice heat capacity C_b of Cr [36] fixes the parameter values in (2) and (3). Values for heat capacities obtained from fitting are within the expected range: the ratio between the lattice and electronic heat capacities during the excitation ($C_b/c_L T \sim 7$) is comparable to that extracted from previous experiments [31] (larger c_L values than in bulk [36] can be explained by disorder in the film [40]), as is the value of thermal coupling constant k [36]. The fitted laser pulse duration ξ agrees with the known laser pulse duration. Absorption at 800 nm wavelength and 30° angle can be estimated based

on measured complex refractive index $n \approx 3 + 1.1i$ for Cr thin films [41] as $\sim 20\%$, while the obtained value of A gives absorption of $Ad/P \sim 15\%$, which is remarkably close considering possible differences due to film roughness. With these fitted parameters, our theoretical model provides an excellent quantitative agreement with the experiment, as Figs. 3 (d, e) demonstrates.”

In summary, this paper will fall in the minor advance in the experiment after the previous paper by the same group, Ref. [8].

We strongly disagree with the reviewer. Our paper [8] involved single pulse measurements and no control. There was no precise theory and model developed in [8], which in this paper was only enabled by the two-pulse measurements.

We hope that with the clarifications we provided with the revised manuscript we alleviate all the reviewers' concerns.